# Disentangling Cognitive Diagnosis with Limited Exercise Labels

**Xiangzhi Chen**[1]    **Le Wu**[1, 3, *]    **Fei Liu**[1]    **Lei Chen**[2]

**Kun Zhang**[1]    **Richang Hong**[1, 3]    **Meng Wang**[1, 4]

[1] Hefei University of Technology    [2] Tsinghua University
[3] Institute of Dataspace, Hefei Comprehensive National Science Center
[4] Institute of Artificial Intelligence, Hefei Comprehensive National Science Center
`{cxz.hfut, lewu.ustc}@gmail.com  feiliu@mail.hfut.edu.cn`
`{chenlei.hfut, zhang1028kun, hongrc.hfut, eric.mengwang}@gmail.com`

## Abstract

Cognitive diagnosis is an important task in intelligence education, which aims at measuring students' proficiency on specific knowledge concepts. Given a fully labeled exercise-concept matrix, most existing models focused on mining students' response records for cognitive diagnosis. Despite their success, due to the huge cost of labeling exercises, a more practical scenario is that limited exercises are labeled with concepts. Performing cognitive diagnosis with limited exercise labels is under-explored and remains pretty much open. In this paper, we propose *Disentanglement based Cognitive Diagnosis (DCD)* to address the challenges of limited exercise labels. Specifically, we utilize students' response records to model student proficiency, exercise difficulty and exercise label distribution. Then, we introduce two novel modules - group-based disentanglement and limited-labeled alignment modules - to disentangle the factors relevant to concepts and align them with real limited labels. Particularly, we introduce the tree-like structure of concepts with negligible cost for group-based disentangling, as concepts of different levels exhibit different independence relationships. Extensive experiments on widely used benchmarks demonstrate the superiority of our proposed model.

## 1 Introduction

In the field of intelligent education systems, cognitive diagnosis (CD) [25, 26] is a fundamental and essential task, which aims at measuring students' proficiency on specific knowledge concepts through the student performance prediction process. CD can serve various applications, such as computerized adaptive testing [2], targeted training [20], and exercise recommendation [24, 14]. We show a toy example of CD in Fig. 1-(a). The exercise-concept matrix (known as Q-matrix) is labeled to indicate which concepts are tested in each exercise. For example, concepts $k_3$ and $k_5$ are tested in exercise $v_3$. The CD model diagnosed that student $u_2$ has a higher proficiency on the two concepts than that of $u_1$ because $u_2$ answered correctly on the exercise while $u_1$ answered incorrectly. It's obvious that fully labeled Q-matrix plays an essential role in interpretability (i.e., diagnostic report) for CD models.

Most existing CD models focus on enhancing the mining process of response records to achieve better diagnosis results. These models typically use a fully labeled Q-matrix that has been annotated by domain experts to train their models. For example, the classical DINA model [8] assumes that students must master all knowledge concepts associated with an exercise to answer it correctly,

---

*Corresponding Author

37th Conference on Neural Information Processing Systems (NeurIPS 2023).

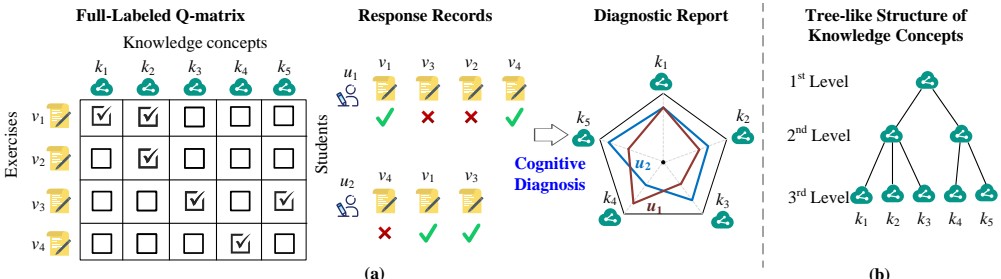

Figure 1: Example of cognitive diagnosis. (a) We show an example to demonstrate the CD process, using a fully-labeled Q-matrix annotated by domain experts. (b) Tree-like structure of concepts is easily annotated with negligible cost as the number of concepts is far fewer than that of exercises.

except in cases where the student guesses correctly. Representative neural-based CD models, such as NCDM [38], KaNCD [39] and KSCD [30], explicitly model the explicit relationship between exercise and knowledge concepts to constrain student proficiency during diagnostic processes. Despite the success of these models, they share a common assumption that the knowledge concepts of each exercise are fully labeled. However, the annotation of exercises is a labor-intensive task that requires the expertise of professionals. There are always many exercises involved in intelligent education systems, and the workload for annotation significantly increases with the number of exercises. Thus, a more practical scenario is cognitive diagnosis with limited exercise labels.

Performing cognitive diagnosis using limited exercise labels is indeed an under-explored area in current research. When faced with the few labeled exercise, a straightforward approach is to prefill the Q-matrix in order to provide a complete Q-matrix as input to an existing CD model. One limitation of this method is that its effectiveness largely depends on the accuracy of the pre-filling algorithm. If the pre-filling algorithm fails to identify the knowledge concepts associated with each exercise correctly, the resulting filled Q-matrix may be inaccurate or incomplete, which can negatively impact the model's diagnostic performance. Therefore, how to achieve interpretable diagnosis on specific knowledge concepts with limited exercise labels remains a challenge.

In this work, we propose *Disentanglement based Cognitive Diagnosis (DCD)* to address the challenges posed by limited exercise labels. Specifically, DCD is a semi-supervised disentanglement method to model the students' proficiency with limited exercise labels. To achieve this, we utilize historical practice records to model student proficiency, exercise difficulty, and exercise relevance on each knowledge concept. Then, we introduce two novel modules - the group-based disentanglement module and the limited-labeled alignment module - to disentangle factors relevant to knowledge concepts and align them with real limited exercise labels. Particularly, we introduce the tree-like structure of concepts (shown in Fig. 1-(b)) for group-based disentangling, as concepts of different levels exhibit different independence relationships. The knowledge concept tree reduces the workload required for annotating knowledge concepts, as there are far fewer concepts than exercises to annotate. These modules work together can deal with the limited exercise labels and diagnose the student's proficiency on each concept with interpretability. The main contributions of this work are summarized as follows:

- Our work represents one of the few attempts to focus on the problem of limited exercise labels in cognitive diagnosis, which is common in real-world practice.

- We propose a semi-supervised disentanglement approach to address the challenge of cognitive diagnosis with limited exercise labels.

- We conduct extensive experiments under different few-labeled settings on three real-world datasets, which demonstrates the effectiveness of our proposed DCD model.

## 2 Related Work

### 2.1 Cognitive Diagnosis in Intelligent Education Systems

The early CD works like IRT [10] and MIRT [35] focus on modeling students' answering process by predicting the probability of a student answering a question correctly, which utilizes latent factors as

the student's ability. IRT and MIRT are also known as Latent Factor Models (LFM). The obvious limitation of the above models is the lack of interpretability, i.e., the inability to obtain an explicit multidimensional diagnostic report on each knowledge concept. To achieve better interpretability, later diagnostic models focus on incorporating knowledge concepts of exercises to diagnose students' proficiency in all knowledge concepts [8, 38, 6, 47, 21]. Representative NCDM [38] adopts neural networks to model non-linear interactions instead of handcrafted interaction functions in previous works [10, 35]. However, researchers find that the free student proficiency vector learning paradigm in NCDM is not capable of tackling weak-knowledge-coverage scenario and model relation among knowledge concepts implicitly to address the problem [39, 30].

**Cognitive Diagnosis in Few-labeled Scenarios.** Most existing CD models are unable to directly deal with the few-labeled scenarios, and assume that the exercises are fully labeled. NCDM+ [39] fills the missing knowledge concepts by TextCNN [18] from the exercise's text information, which regards it as a previous task for CD. Therefore, the performance of CD lies in the accuracy of pre-filled knowledge concepts. In this work, we focus on a more challenging scenario of missing exercise textual information and infer missing knowledge concepts from response records only.

## 2.2 Disentangled Representation Learning

Disentangled Representation Learning (DRL) is an important learning paradigm that aims to disentangle the underlying generative factors hidden in the observed data into the latent variables in the representation [1]. One notable benefit attributed to DRL is its ability to extract latent factors that embody semantic meanings, thus enhancing the interpretability of machine learning models. Both VAE [19] and GAN [12] based DRL are widely adopted techniques. Considering that the training of VAE is more stable than that of GAN [17, 45], in this work, we extend the VAE framework and focus on learning interpretable disentangled representations to help cognitive diagnosis.

**VAE-based Disentangled Representation Learning.** $\beta$-VAE [13] modifies the Evidence Lower Bound (ELBO) of VAE [19] by adding a hyperparamter $\beta$ on KL term and find that a larger $\beta$ corresponds to the better disentanglement of latent code. However, the increasing $\beta$ would also lead to the worse reconstruction error dramatically [17, 36]. To analyze the trade-off between reconstruction accuracy and the quality of disentangled representation, both FactorVAE [17] and $\beta$-TCVAE [5] show that the success of $\beta$-VAE in learning disentangled representations can be attributed to penalizing the Total Correlation (TC) term. $\beta$-TCVAE decomposes the expected KL term in ELBO into index-code MI term, TC term, and dimension-wise KL term. The TC term would decrease the mutual information among factors and improve independence among latent factors, which results in better disentanglement. However, some researchers claim that the above unsupervised learning of disentangled representations is fundamentally impossible without inductive biases as sometimes the target dataset is not semantically clear and well-structured to be disentangled [27]. Hence, more works focus on weak-supervised approaches [28, 37] and semi-supervised [29, 16]. A theoretical framework is provided to assist in analyzing the disentanglement guarantees by weak supervision methods (e.g. restricted labeling, match pairing, and rank pairing) [37]. Some researchers validate that with little and imprecise supervision (e.g. manual labeling of factors) it is possible to reliably learn disentangled representations [29]. In this work, we regard limited exercise labels as a semi-supervision to help disentangle knowledge concepts-related factors.

**Disentangled Representation Learning for User Modeling.** User modeling [22] aims to capture a number of attributes of each user, with the help of items, item features and/or user-item response matrix [43], etc. DRL has a wide range of applications in user modeling to disentangle attributes. For example, recommendation with several aspects of users' interests [23, 31, 46, 40], fair user representation to disentangle sensitive attributes [7, 34].

## 3 Problem Formulation

Let $\mathcal{U} = \{u_1, u_2, \ldots, u_N\}$, $\mathcal{V} = \{v_1, v_2, \ldots, v_M\}$, and $\mathcal{K} = \{k_1, k_2, \ldots, k_K\}$ denote the sets of students, exercises, and knowledge concepts, respectively, where $N$, $M$, and $K$ represent the size of each set. The response records are denoted as $\mathbf{X} = \{X_{ij}\}_{N \times M}$, where $X_{ij}$ equals 1, 0 or -1 representing that the student answered the exercise correctly, incorrectly, or did not answer the exercise, respectively. The relationship between exercises and knowledge concepts is represented by the Q-matrix $\mathbf{Q} = \{Q_{ij}\}_{K \times M}$. Typically, each exercise is related to multiple knowledge concepts.

$Q_{ij}$ equals 1 or 0 representing that exercise $v_j$ is related to the knowledge concept $k_i$ or not. In this work, we explore the few-labeled situation, that is, there are only a few exercises that are labeled. Thus, we split exercise set $\mathcal{V}$ into the labeled set $\mathcal{V}_1$ and the unlabeled set $\mathcal{V}_2$.

There are correlations and independence among different knowledge concepts. In our work, we introduce the knowledge concept tree [41] to model their relations at different levels. By utilizing this tree structure, we can effectively capture knowledge concepts' interrelationships. For convenience, we transform the knowledge concept tree into a standard tree (the height of all subtrees of any node in the tree is equal). For more details about knowledge concept tree, please see Appendix C. Suppose we construct a standard tree $G$ with $L$ levels, wherein its leaf nodes are denoted by the set $\mathcal{K} = \{k_1, k_2, \ldots, k_K\}$. Each level of knowledge concept can be regarded as a grouping of the last level of knowledge concepts. The deeper the level, the finer the granularity of the grouping. Let $G^i = \{G_1^i, G_2^i, \ldots, G_{|G^i|}^i\}$ denote grouping method of the $i$-th level knowledge concepts, where $i = 1, 2, \ldots, L$ and $\left|G^{i+1}\right| \geq \left|G^i\right|$. Therefore, $G^1 = \{G_1^1 = \mathcal{K}\}$ and $G^L = \{G_1^L = \{k_1\}, G_2^L = \{k_2\}, \ldots, G_K^L = \{k_K\}\}$.

**Problem Definition.** In an intelligent education system, given student set $U$, exercise set $\mathcal{V} = \mathcal{V}_1 \cup \mathcal{V}_2$, knowledge concepts $\mathcal{K}$, response records $\mathbf{X}$, knowledge concept tree $G$ and incomplete Q-matrix, our goal is to diagnose students' cognitive states on knowledge concept set $\mathcal{K}$ with few-labeled exercises, and predict the scores of students doing exercises.

# 4 The Proposed Model

In cognitive diagnosis, it is crucial to determine whether a student has the proficiency in knowledge concepts required to answer an exercise correctly. This becomes even more challenging when there are limited exercise labels available. To model students' proficiency with few-labeled exercises, we design disentanglement and alignment modules that can effectively utilize the weak supervision of a few labeled exercises.

The overall framework of DCD is illustrated in Fig. 2, including input, encoder, disentanglement, alignment, and decoder. The input of DCD contains the student-exercise interaction matrix $\mathbf{X}$ and the few-labeled matrix $\mathbf{Q}$. For diagnosing students' cognitive states, we first utilize the encoder module to model three important components from $\mathbf{X}$: student proficiency, exercise difficulty, and exercise relevance. Then, we leverage the group-based disentanglement module to separate the factors related to knowledge concepts from the three components mentioned above. Next, the limited-labeled alignment module associates each decoupled factor with a knowledge concept semantic with the missing matrix $\mathbf{Q}$. Finally, the decoder module predicts the scores based on the student proficiency, exercise difficulty, and exercise relevance. Overall, DCD provides an effective approach for cognitive diagnosis under limited exercise labels.

## 4.1 Encoder Module

The interaction matrix $\mathbf{X}$ provides valuable information regarding student proficiency, exercise difficulty, and exercise relevance from different perspectives. Specifically, 1) From a student-centric (row-wise) perspective, the response records of a student reflect his proficiency on all knowledge concepts. 2) From an exercise-centric (column-wise) perspective, the response records of an exercise reflect exercise difficulty and exercise relevance on all knowledge concepts. On the one hand, to effectively model from two perspectives, we design the student proficiency encoder, exercise difficulty encoder, and exercise relevance encoder modules. On the other hand, following existing cognitive models, an interaction function (corresponding to our decoder) to predict the answering result with student traits and exercise traits as input (corresponding to our encoders), which ensures the interpretability of student proficiency representation. Traditional single encoder is not applicable for cognitive diagnosis.

**Student Encoder.** The student encoder module is intended to infer student's proficiency on each knowledge concept from $\mathbf{X}$, which models true posterior distribution $p(\mathbf{z}_u|\mathbf{x}_u)$ by constructing an inference network $f_{\phi_u}(\mathbf{x}_u)$ corresponding to approximate posterior distribution $q_{\phi_u}(\mathbf{z}_u|\mathbf{x}_u)$, parameterized by $\phi_u$. We assume student's proficiency on all knowledge concepts $\mathbf{z}_u$ follows a multivariate standard Gaussian distribution prior, i.e., $p(\mathbf{z}_u) \sim \mathcal{N}(\mathbf{0}, \mathbf{I})$. The approximate posterior

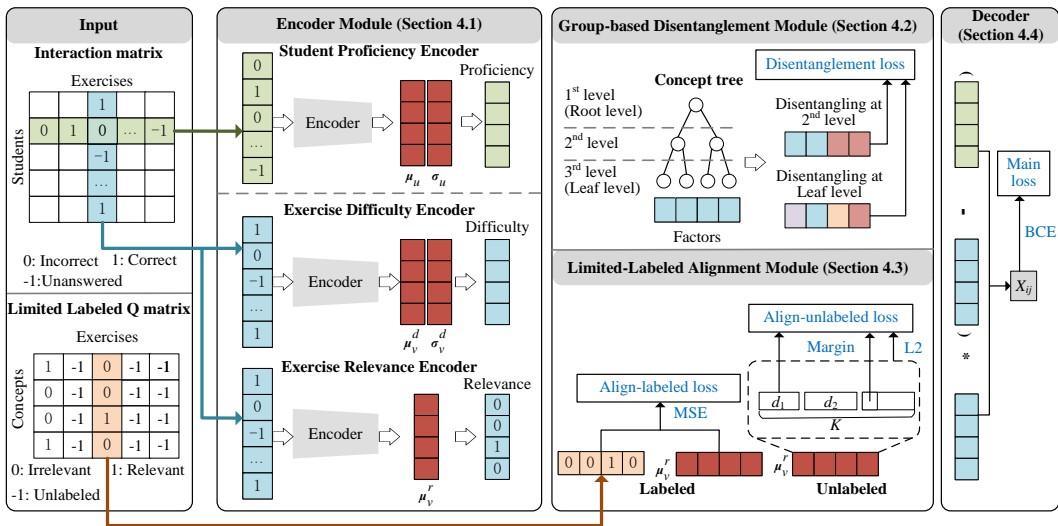

Figure 2: The overall framework of the proposed DCD. We first encode student proficiency, exercise difficulty, and relevance vectors from students' historical response records. Then, we devise two novel modules ( i.e., the group-based disentanglement and the limited-labeled alignment modules) to disentangle factors relevant to knowledge concepts and align them with real limited exercise labels. Finally, the decoder module predicts the scores based on the student proficiency, exercise difficulty, and exercise relevance.

for $K$-dimensional latent code $\mathbf{z}_u \in \mathbb{R}^K$ is expressed as follows:

$$q_{\phi_u}(\mathbf{z}_u|\mathbf{x}_u) = \prod_{k=1}^{K} \mathcal{N}(\mathbf{z}_u[k]; \boldsymbol{\mu}_u[k], \boldsymbol{\sigma}_u[k]), \qquad (1)$$

where $\boldsymbol{\mu}_u \in \mathbb{R}^K$ and $\boldsymbol{\sigma}_u \in \mathbb{R}^K_{>=0}$ are generated from student encoder $f_{\phi_u}(\mathbf{x}_u)$. The $[\cdot]$ indicates the index selection operation.

**Exercise Encoders.** We design two encoders at exercise side: exercise difficulty and exercise relevance encoders. The two encoders aim to infer exercise's difficulty and exercise's relevance on each knowledge concept from $\mathbf{X}$, respectively. For exercise difficulty encoder, we assume exercise's difficulty on all knowledge concepts $\mathbf{z}_v^d$ follows a multivariate standard Gaussian distribution prior similar to the student encoder, i.e., $p(\mathbf{z}_v^d) \sim \mathcal{N}(\mathbf{0}, \mathbf{I})$.

We assume exercise's relevance on all knowledge concepts $\mathbf{z}_v^r$ follows a multivariate Bernoulli prior, i.e. $p(\mathbf{z}_v^r) \sim Bernoulli(\mathbf{p})$, where $\mathbf{p}$ is a hyperparameter. The exercise relevance encoder models true posterior distribution $p(\mathbf{z}_v^r|\mathbf{x}_v)$ by constructing an inference network $f_{\phi_v^r}(\mathbf{x}_v)$ corresponding to approximate posterior distribution $q_{\phi_v^r}(\mathbf{z}_v^r|\mathbf{x}_v)$, parameterized by $\phi_v^r$. The approximate posterior for K-dimensional latent code $\mathbf{z}_v^r \in \{0,1\}^K$ is expressed as:

$$q_{\phi_v^r}(\mathbf{z}_v^r|\mathbf{x}_v) = \prod_{k=1}^{K} Bernoulli(\mathbf{z}_v^r[k]; \boldsymbol{\mu}_v^r[k]), \qquad (2)$$

where $\boldsymbol{\mu}_v^r \in [0,1]^K$ are generated from exercise encoder $f_{\phi_v}(\mathbf{x}_v)$.

## 4.2 Group-based Disentanglement Module

The representations obtained by the encoder module do not relates to the knowledge concepts, which makes it difficult to perform an accurate cognitive diagnosis. To address this issue, we assume the entire response records as evolving from three groups of generative factors: student proficiency factors on each knowledge concept, exercise difficulty factors on each knowledge concept, and exercise relevance factors on each knowledge concept. We desire to disentangle these factors following existing DRL methods.

Inspired by the outstanding performance of $\beta$-TCVAE [5] in disentanglement, we introduce the structure of $\beta$-TCVAE to obtain a better trade-off between the reconstruction accuracy and the quality of disentangled representation. We find the index-code term of $\beta$-TCVAE plays a side effect on performance and drop it during optimization like [32, 7]. Take student proficiency factors as an example, the eventual Totoal Correlation (TC) constraint on $\mathbf{z}_u$ can be expressed as follows:

$$\mathcal{L}_d(\mathbf{z}_u) = D_{KL}(q(\mathbf{z}_u)||\prod_{i=1}^{K} q(\mathbf{z}_u[i])), \tag{3}$$

where $q(\mathbf{z}_u) = \sum_{i=1}^{N} q(\mathbf{z}_u|\mathbf{x}_{u_i})p(\mathbf{x}_{u_i})$ is the aggregated posterior of $\mathbf{z}_u$. The TC term would decrease the mutual information among latent factors and improve independence among latent factors. The dimension-wise KL constraint on $\mathbf{z}_u$ can be written as follows:

$$\mathcal{L}_p(\mathbf{z}_u) = \sum_{i=1}^{K} D_{KL}(q(\mathbf{z}_u[i])||p(\mathbf{z}_u[i])), \tag{4}$$

where $p(\mathbf{z}_u[i])$ follows the prior standard Gaussian distribution. The dimension-wise KL term prevents each latent variable from deviating too far from specified priors.

However, some CD works argue that the knowledge concepts are not independent and employ relation among knowledge concepts to improve CD [11, 39]. The direct independence constraint for each knowledge concept may be not applicable for cognitive diagnosis, which is also verified in our experiment. To address the challenge, we propose to utilize the easy-labeled knowledge concept tree to capture underlying independence among knowledge concepts. We assume there is less independence among groups of knowledge concepts grouped by their parent nodes in the knowledge concept tree. Corresponding to the real world, knowledge concepts in the same chapter would be related more than knowledge concepts in different chapters. Therefore, each level concept in the tree is a kind of grouping method for the last level concepts. The deeper the level, the finer the granularity of the grouping. Considering the grouping methods at the $i$-th level, the Eq. (3) can be rewritten as follows:

$$\mathcal{L}_d^i(\mathbf{z}_u) = D_{KL}(q(\mathbf{z}_u)||\prod_{j=1}^{|G^i|} q(\mathbf{z}_u[G_j^i])), \tag{5}$$

where $G_j^i$ is the knowledge concepts that belong to $j$-th group according to the grouping method of $i$-th level in the tree. The Eq. (5) allows dependence in intra-group knowledge concepts and maintains independence among inter-group knowledge concepts.

### 4.3 Limited-Labeled Alignment Module

After applying the group-based disentanglement module, it is important to correspond each latent factor with an actual knowledge concept to ensure the interpretability. However, there are only a few labeled exercises with knowledge concepts, making it challenging to perform the accurate alignment. Inspired by semi-supervised DRL, to address this issue, we devised an alignment module that aligns both few labeled and numerous unlabeled exercises separately. For convenience, we use $\mathbf{q}_j \in \{0,1\}^K$ to denote $j$-th column of matrix $\mathbf{Q}$ corresponding to exercise $v_j$ that belongs to $\mathcal{V}_1$.

**Few Labeled Exercises.** For an exercise $v_j \in \mathcal{V}_1$, we utilize the Mean Square Error (MSE) loss to constraint similarity between $\mathbf{q}_j$ and $\boldsymbol{\mu}_{v_j}^r$, which can be defined as follows:

$$\mathcal{L}_l = \sum_{v_j \in \mathcal{V}_1} ||\mathbf{q}_j - \boldsymbol{\mu}_{v_j}^r||_2^2, \tag{6}$$

where $\mathcal{V}_1$ means labeled exercise set. For semi-supervised DRL in the vision field, it's usually enough with a single alignment module for a few labeled samples. However, for cognitive diagnosis, the annotation vector (i.e., the column of Q-matrix) of exercise is very sparse with only one or two knowledge concepts per exercise. The challenge is that utilizing a few labeled exercises to do alignment is too hard to keep such sparsity in relevance representation for unlabeled exercises. To this end, the alignment of unlabeled exercises is necessary.

**Numerous Unlabeled Exercises.** For an exercise $v_j \in \mathcal{V}_2$, we have no idea any knowledge concept of it. But we know there are only a few knowledge concepts of each exercise. The feature of sparsity

in $\boldsymbol{\mu}^r_{v_j}$ would help infer missing knowledge concepts. We just need to guarantee that there are only a few elements to be one and other elements to be zero in $\boldsymbol{\mu}^r_{v_j}$. In this work, we employ a margin loss and an L2 loss to achieve this target, which can be expressed as follows:

$$\mathcal{L}_{ul} = \sum_{v_j \in \mathcal{V}_2} (\lambda_1 max(0, m - (\boldsymbol{\mu}^r_{v_j}[max\#d_1] - \boldsymbol{\mu}^r_{v_j}[max@(d_1 + d_2 + 1)])) + \lambda_2 \|\boldsymbol{\mu}^r_{v_j}\|^2_2), \quad (7)$$

where $m$ is the margin, $\lambda_1$ is the hyperparameter to control margin loss, and $\lambda_2$ is the hyperparameter to control L2 loss. We use $\boldsymbol{\mu}^r_v[max@k]$ and $\boldsymbol{\mu}^r_v[max\#k]$ to denote the $k$-th largest element of $\boldsymbol{\mu}^r_v$ and the top $k$ largest elements of $\boldsymbol{\mu}^r_v$, respectively. In Eq. (7), we first sort $\boldsymbol{\mu}^r_{v_j}$ by descending order and then split the sorted one into three parts of length $d_1, d_2, K - d_1 - d_2$. We argue that knowledge concepts in the last part are the least likely to be the ground-truth knowledge concepts of exercise $v_j$ and knowledge concepts in the first part are the most likely to be the ground-truth knowledge concepts of exercise $v_j$. Therefore, we adopt a margin loss between knowledge concepts in the first part and knowledge concepts in the last part, which requires that all values in the first part (i.e., $\boldsymbol{\mu}^r_{v_j}[max\#d_1]$) is at least m greater than the largest value in the last one (i.e., $\boldsymbol{\mu}^r_{v_j}[max@(d_1 + d_2 + 1)]$). For the middle part, we do not constrain anything on it. Intuitively, it's reasonable that adopt an L1 loss to constraint the value in the last part (i.e., $\boldsymbol{\mu}^r_{v_j}[min\#(K - d_1 - d_2)]$) to be 0. In our initial attempt, we first directly try to use L1 loss for numerous unlabeled exercises, and we find it does not show competing performance (detailed in Appendix A.3). We speculate a possible reason is as follows: with L1 loss, after model initialization, it is very hard to revise these incorrectly labeled exercises. In contrast, L2 loss are sensitive to outliers, and would have a larger loss when the corresponding knowledge is incorrectly predicted.

## 4.4   Decoder Module

After aligning the exercise relevance representation with the real knowledge concepts, the decoder module predicts scores of students doing exercises, which could be described as a triplet set $T = \{(u_i, v_j, X_{ij})|X_{ij} \neq -1\}$. The object function can be described as follows:

$$\begin{aligned}
\mathcal{L}_m &= -\mathbb{E}_{q_{\phi_u}(\mathbf{z}_u|\mathbf{x}_u), q_{\phi^d_v}(\mathbf{z}^d_v|\mathbf{x}_v), q_{\phi^r_v}(\mathbf{z}^r_v|\mathbf{x}_v)} [\log p(\mathbf{X}|\mathbf{z}_u, \mathbf{z}^d_v, \mathbf{z}^r_v)] \\
&= \sum_{(u_i, v_j, X_{ij}) \in T} BCE(\sigma((\mathbf{z}_{u_i} - \mathbf{z}^d_{v_j}) \otimes \mathbf{z}^r_{v_j}), X_{ij}),
\end{aligned} \quad (8)$$

where $BCE(\cdot, \cdot)$ is the binary cross entropy loss function, $\otimes$ and $\sigma(\cdot)$ denote inner product operation and sigmoid function, respectively.

Similar to existing CD works [38, 39, 21], the interaction function (i.e., the decoder) keeps the monotonicity assumption [39], which guarantees the interpretability of student proficiency representation (i.e., the higher the value of the latent factor means the better proficiency on the corresponding knowledge concept, detailed in Appendix A.4). In addition, the element-wise operation in the decoder module propagates the alignment (i.e., each latent factor corresponds to each real knowledge concept) on exercise relevance representation to both student proficiency representation and exercise difficulty representation, which is also the same as existing works.

**Optimization.** We summarized all object functions in all modules and combine them for final optimization. We set different hyperparameters to balance each loss function. The final object function can be summarized as follow:

$$\underset{\Theta=[\phi_u, \phi^d_v, \phi^r_v]}{\arg\min} \mathcal{L} = \mathcal{L}_m + \alpha\mathcal{L}_l + \mathcal{L}_{ul} + \sum_{i=1}^{L}(\beta^i \sum_{\mathbf{z} \in \{\mathbf{z}_u, \mathbf{z}^d_v, \mathbf{z}^r_v\}} \mathcal{L}^i_d(\mathbf{z})) + \sum_{\mathbf{z} \in \{\mathbf{z}_u, \mathbf{z}^d_v, \mathbf{z}^r_v\}} \mathcal{L}_p(\mathbf{z}), \quad (9)$$

where $\Theta = [\phi_u, \phi^d_v, \phi^r_v]$ is the parameter set in the whole model, $\alpha$ is the hyperparameter for alignment of labeled exercises, and $\beta^i$ denotes the weight for disentanglement term corresponding to the $i$-th level in the knowledge concept tree.

# 5 Experiments

## 5.1 Experimental Settings

**Datasets.** Our experiments are conducted on three real-world datasets, i.e., Matmat[2], Junyi [4] and NIPS2020EC [42], all of which contain knowledge concepts of the tree structure. The statistics of these datasets are summarized in Table 1. We adopt five-fold cross-validation to avoid randomness. The details about datasets and implementation[3] are depicted in the Appendix B.

Table 1: The statistics of three datasets.

| dataset | #students | #exercises | #concepts at different level | #leaf concepts per exercise | #right records : #error records | sparsity |
|---------|-----------|------------|------------------------------|------------------------------|----------------------------------|----------|
| Matmat | 7,067 | 1,847 | [1,5,11] | 1.00 | 5.50 | 97.7% |
| Junyi | 8,852 | 720 | [1,8,39] | 1.00 | 2.16 | 87.4% |
| NIPS2020EC | 11,857 | 6,509 | [1,4,29,64] | 1.05 | 2.00 | 99.1% |

**Metrics.** To evaluate the effectiveness of DCD in terms of score prediction and interpretability, we adopt three widely used prediction metrics: AUC [3], ACC, RMSE, and one interpretability metric: Degree of Agreement (DOA) [38]. DOA is the most commonly used interpretability metric in the CD, which measures the degree of agreement between cognitive results and response records.

**Baselines.** The baselines including the classical data mining model Probabilistic Matrix Factorization (PMF) [33], the latent factor models IRT [10], MIRT [35], and the representative CD models DINA [8], NCDM [38], KaNCD [39] and KSCD [30]. In addition, we also explore a simplified version of our method. Specifically, we remove the exercise difficulty module and margin loss, and focus solely on the fine-grained disentanglement module. This simplified version serves as a baseline, which we refer to as $\beta$-TCVAE in the following analysis.

Existing non-interpretable models such as PMF, IRT, and MIRT cannot obtain a student's proficiency for each knowledge concept [38]. Consequently, DOA cannot be calculated for these models. Interpretable models such as DINA, NCDM, KaNCD, and KSCD cannot directly apply to few-labeled scenarios. To ensure fairness in comparison, we propose a pre-filling algorithm (detailed in Appendix B.2) for the missing Q-matrix before applying these models.

## 5.2 Performance Comparison

To evaluate the effectiveness of our proposed DCD model in few-labeled scenarios, we have conducted experiments on datasets with 10% Q-matirx (few-labeled exercises), 20% Q-matirx (few-labeled exercises) and 100% Q-matirx (fully-labeled exercises). The detailed performance comparison on three datasets is displayed in Table 2.

**Prediction Comparison.** 1) Our proposed model consistently outperforms all interpretable baseline models across the different scenarios, demonstrating its effectiveness. DINA, which only considers binary space, performs the worst. Both KSCD and KaNCD implicitly model relation among knowledge concepts and has a better performance than NCDM. 2) However, all non-interpretable models achieve the best prediction metrics over interpretable models on Junyi and NIPS2020EC datasets except Matmat dataset. This may be because there are far fewer knowledge concepts in the Matmat dataset than in the other two datasets. To meet the interpretability of larger knowledge concept space, interpretable models lose partial prediction accuracy. This explanation is supported by the fact that the prediction results of interpretable models show a noticeable improvement in the few-labeled scenario compared to the fully labeled scenario, indicating that the trade-off between interpretability and accuracy is more significant in a larger knowledge concept space.

**Interpretability Comparison.** 1) Table 2 demonstrates that our method achieves the best interpretability in all scenarios. Especially, for the 10% and 20% Q-matrix (few-labeled scenarios), our model significantly outperforms interpretable baselines in terms of DOA, demonstrating its superior interpretability in this scenario. 2) In few-labeled scenarios, KaNCD and KSCD show a remarkable improvement in DOA compared to DINA and NCDM. This is because DINA and NCDM can only

---

[2]`https://github.com/adaptive-learning/matmat-web`
[3]Our code is available at `https://github.com/kervias/DCD`

Table 2: Comparison in both fully-labeled and few-labeled scenarios. We divide models into non-interpretable and interpretable models. The best scores are in bold for two kinds of models. Noted that non-interpretable models keep same results in any scenario as they do not utilize Q-matrix.

| Model | 100% Q-matrix (fully-labeled) | | | | 20% Q-matrix (few-labeled) | | | | 10% Q-matrix (few-labeled) | | | |
|---|---|---|---|---|---|---|---|---|---|---|---|---|
| | AUC↑ | ACC↑ | RMSE↓ | DOA↑ | AUC↑ | ACC↑ | RMSE↓ | DOA↑ | AUC↑ | ACC↑ | RMSE↓ | DOA↑ |
| Matmat | | | | | | | | | | | | |
| PMF | **0.759** | **0.857** | **0.334** | - | **0.759** | **0.857** | **0.334** | - | **0.759** | **0.857** | **0.334** | - |
| IRT | 0.747 | 0.851 | 0.339 | - | 0.747 | 0.851 | 0.339 | - | 0.747 | 0.851 | 0.339 | - |
| MIRT | 0.750 | 0.853 | 0.337 | - | 0.750 | 0.853 | 0.337 | - | 0.750 | 0.853 | 0.337 | - |
| DINA | 0.696 | 0.816 | 0.368 | 0.828 | 0.698 | 0.774 | 0.380 | 0.766 | 0.700 | 0.770 | 0.372 | 0.735 |
| NCDM | 0.742 | 0.839 | 0.348 | 0.850 | 0.738 | 0.837 | 0.350 | 0.781 | 0.740 | 0.846 | 0.346 | 0.738 |
| $\beta$-TCVAE | 0.753 | 0.847 | 0.337 | 0.847 | 0.733 | 0.843 | 0.342 | 0.743 | 0.730 | 0.842 | 0.343 | 0.733 |
| KSCD | 0.753 | 0.824 | 0.348 | 0.833 | 0.745 | 0.828 | 0.348 | 0.789 | 0.744 | 0.838 | 0.345 | 0.769 |
| KaNCD | 0.760 | **0.857** | 0.335 | 0.860 | 0.752 | 0.852 | 0.338 | 0.800 | 0.751 | 0.852 | 0.338 | 0.783 |
| **DCD** | **0.763** | **0.857** | **0.334** | **0.861** | **0.767** | **0.857** | **0.333** | **0.819** | **0.764** | **0.855** | **0.334** | **0.796** |
| Junyi | | | | | | | | | | | | |
| PMF | 0.817 | 0.776 | 0.394 | - | 0.817 | 0.776 | 0.394 | - | 0.817 | 0.776 | 0.394 | - |
| IRT | **0.822** | **0.779** | **0.391** | - | **0.822** | **0.779** | **0.391** | - | **0.822** | **0.779** | **0.391** | - |
| MIRT | 0.820 | 0.777 | 0.392 | - | 0.820 | 0.777 | 0.392 | - | 0.820 | 0.777 | 0.392 | - |
| DINA | 0.737 | 0.684 | 0.462 | 0.862 | 0.748 | 0.658 | 0.459 | 0.638 | 0.754 | 0.663 | 0.456 | 0.599 |
| NCDM | 0.760 | 0.715 | 0.445 | 0.851 | 0.791 | 0.743 | 0.418 | 0.632 | 0.804 | 0.753 | 0.409 | 0.594 |
| $\beta$-TCVAE | 0.770 | 0.732 | 0.423 | 0.827 | 0.803 | 0.764 | 0.406 | 0.699 | 0.805 | 0.763 | 0.407 | 0.673 |
| KSCD | 0.767 | 0.725 | 0.426 | 0.809 | 0.804 | 0.754 | 0.409 | 0.670 | 0.809 | 0.760 | 0.405 | 0.650 |
| KaNCD | 0.775 | **0.754** | 0.422 | 0.835 | 0.807 | 0.767 | **0.400** | 0.719 | **0.815** | **0.773** | **0.395** | 0.691 |
| **DCD** | **0.787** | **0.754** | **0.414** | **0.873** | **0.811** | **0.768** | **0.400** | **0.733** | 0.814 | 0.771 | 0.397 | **0.717** |
| NIPS2020EC | | | | | | | | | | | | |
| PMF | 0.814 | 0.760 | 0.407 | - | 0.814 | 0.760 | 0.407 | - | 0.814 | 0.760 | 0.407 | - |
| IRT | 0.819 | 0.762 | 0.402 | - | 0.819 | 0.762 | 0.402 | - | 0.819 | 0.762 | 0.402 | - |
| MIRT | **0.822** | **0.765** | **0.400** | - | **0.822** | **0.765** | **0.400** | - | **0.822** | **0.765** | **0.400** | - |
| DINA | 0.764 | 0.705 | 0.451 | 0.856 | 0.767 | 0.687 | 0.453 | 0.740 | 0.766 | 0.684 | 0.453 | 0.693 |
| NCDM | 0.795 | 0.730 | 0.423 | 0.853 | 0.803 | 0.751 | 0.411 | 0.736 | 0.805 | 0.750 | 0.411 | 0.687 |
| $\beta$-TCVAE | 0.792 | 0.744 | 0.417 | 0.854 | 0.797 | 0.740 | 0.416 | 0.781 | 0.801 | 0.745 | 0.413 | 0.774 |
| KSCD | 0.798 | 0.716 | 0.421 | 0.830 | 0.809 | 0.754 | 0.410 | 0.787 | 0.809 | 0.755 | 0.410 | 0.777 |
| KaNCD | 0.797 | 0.750 | 0.423 | 0.856 | 0.811 | 0.759 | 0.403 | 0.783 | 0.812 | 0.760 | **0.404** | 0.761 |
| **DCD** | **0.801** | **0.752** | **0.415** | **0.861** | **0.812** | **0.761** | **0.405** | **0.793** | **0.813** | **0.762** | **0.404** | **0.786** |

diagnose knowledge concepts labeled in the training set. When incorrect knowledge concepts are filled in, KaNCD and KSCD are less affected, while DINA and NCDM cannot accurately diagnose the true knowledge concepts. In contrast, our model avoids this issue by modeling the distribution of students' proficiency overall knowledge concepts.

## 5.3 Disentanglement Analysis

To investigate the effectiveness of disentanglement, we conducted an experiment where we compared our method at different disentanglement weights $\beta$ in the few-labeled scenario. Fig. 3 (a-d) illustrates the results, the x-axis shows the value of $\beta$ from 0 to 2, which corresponds to the strength of disentanglement. The AUC is displayed on the left y-axis and the DOA is displayed on the right y-axis. The AUC(i) and DOA(i) denote the results by disentanglement only according to i-th level of knowledge concept tree. There are some observations from Fig. 3: 1). The disentanglement according to the last level of tree (i.e., the most fine-grained disentanglement) can not have an improvement in terms of DOA as the rise of $\beta$. The phenomenon is consistent with the assumption that knowledge concepts are correlated. Despite there being an improvement in AUC, we assume it's the result of interpretability losses, as mentioned in the prediction comparison in section 5.2. 2) The coarse-grained disentanglement could simultaneously improve the AUC and DOA compared to no disentanglement. We observe that the performance increased initially with the implementation of disentanglement and then decreased as the strength of disentanglement became excessive, leading to counterproductivity.

## 5.4 Alignment Analysis

To evaluate the effectiveness of the limited-labeled alignment module, we conducted an experiment with different hyperparameters on Junyi dataset with 10% Q-matrix, namely $\alpha$, $\lambda_1$, $\lambda_2$ and $d_2$, as shown in Fig. 3 (e-h). Due to the number of leaf concepts per exercise being close to 1 for all datasets,

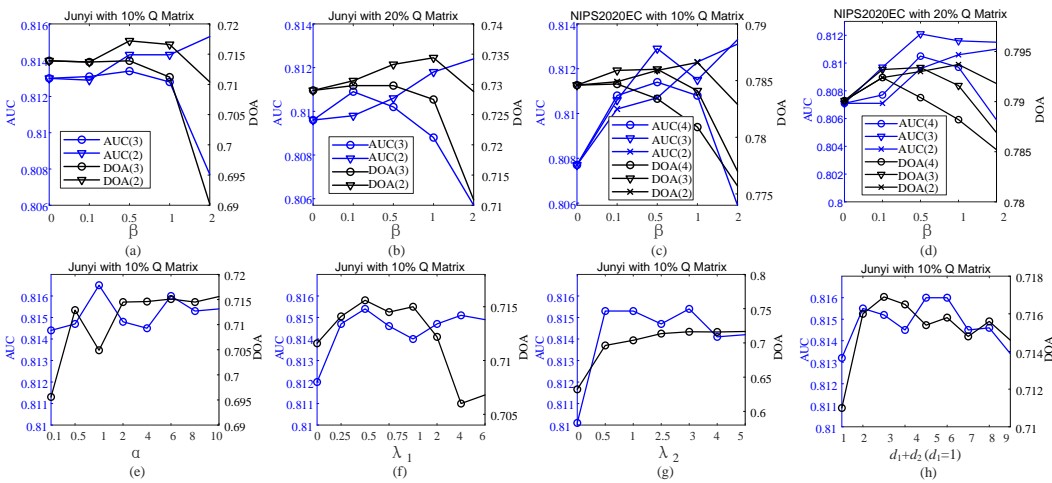

Figure 3: AUC and DOA results of DCD with hyper-parameters in the group-based disentanglement (a-d) and limited-labeled alignment (e-h) modules.

we set the $d_1 = 1$ as the default value. The default margin is 0.5. There are some observations: 1) $\alpha$ is the weight of the alignment loss corresponding to labeled exercises. Fig. 3 (e) shows that the performance of our model initially improves and eventually stabilizes as $\alpha$ increases. 2) $\lambda_1$ is the weight of margin loss for unlabeled exercises. As illustrated in Fig. 3 (f), the employment of margin loss is effective according to results from zero to non-zero of $\lambda_1$. The excessive $\lambda_1$ would decrease the performance. 3) $\lambda_2$ is the weight of L2 loss of unlabeled exercises exercise relevance representation. We can observe that the employment of the L2 loss plays an essential role from Fig. 3 (g). 4) We can observe that the employment of $d_2$ is effective from Fig. 3 (h). As $d_2$ increases from 0 to 8, the performance increases first and then drops. Overall, the limited-labeled alignment module has a positive effect on performance, and the hyperparameters of the alignment module need to be certainly adjusted.

## 5.5 Ablation Study

As shown in Table 3, to prove the effectiveness of our proposed disentanglement and alignment modules, we conduct the ablation study on Junyi dataset under the 10% Q-matrix scenario. The margin Loss in alignment module ensures few elements to be one in exercise relevance repre-

Table 3: Ablation study of DCD on Junyi.

| Model | 10% Q-matrix(few-labeled) | | | |
|---|---|---|---|---|
| | AUC↑ | ACC↑ | RMSE↓ | DOA↑ |
| DCD w/o Alignment-Margin | 0.8140 | 0.7692 | 0.3976 | 0.7136 |
| DCD w/o Alignment-L2 loss | 0.8101 | 0.7681 | 0.3977 | 0.6322 |
| DCD w/o Disentanglement | 0.8130 | 0.7682 | 0.3983 | 0.7134 |
| DCD | **0.8143** | **0.7713** | **0.3965** | **0.7174** |

sentation, which achieves an improvement of 0.53% in terms of DOA. The L2 loss in alignment module ensures the sparsity of exercise relevance representation, which achieves an improvement of 13.48% in terms of DOA. It proves that utilizing a few labeled exercises to do alignment is too hard to keep such sparsity in relevance representation for unlabeled exercises. The disentanglement module alone achieves 0.56% in terms of DOA. Overall, both the proposed disentanglement and alignment modules achieve an improvement in terms of prediction and interpretability metrics.

## 6 Conclusion

In this paper, we present a novel approach, called DCD, to tackle the interpretability problem of cognitive diagnosis with limited exercise labels. Inspired by semi-supervised DRL, we learn disentangled representations and align them with real limited labels by group-based disentanglement and limited-labeled alignment. Extensive experiments on widely used benchmarks demonstrate the superiority of our proposed method in few-labeled scenarios.

## Acknowledgments and Disclosure of Funding

This work was supported in part by grants from the National Key Research and Development Program of China (Grant No. 2021ZD0111802), the National Natural Science Foundation of China (Grant No. 61972125, 72188101, 61932009, 62376086, and U22A2094).

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

## A  Additional Experiments

### A.1  Comparison on Inferring Missing Knowledge Concepts

To evaluate the degree of alignment from exercises' relevance, we adopt Hit Ratio (HR) [44] to measure whether the top-k ranked elements in exercise relevance representation hit a ground-truth knowledge concept. The algorithm about filling missing Q-matrix for baselines (detailed in Appendix B.2) is compared with our exercise relevance encoder. The filling results of HR@2 is illustrated in Table 4. We can observe that: 1) It's rational that inferring missing knowledge concepts according to response records of exercises. The filling algorithm and our method show a obvious improvement than randomly assigning knowledge concepts for unlabeled exercises. 2) Our method demonstrates the best HR@2 on Matmat dataset, but shows a worse performance on other datasets. This is because the number of knowledge concepts in Matmat has a smaller scale than other datasets and our model has a better inferring performance in small scale knowledge concepts. Despite the worse filling results than filling algorithm for baselines, our method still shows the best performance in prediction metrics and interpretable metrics, which demonstrate the robustness of the proposed DCD approach. This may be attributed to the inference of student proficiency of our student encoder module.

Table 4: HR@2 comparison on inferring missing knowledge concepts

|  | Matmat | | Junyi | | NIPS2020EC | |
|---|---|---|---|---|---|---|
|  | 20% Q-matrix | 10% Q-matrix | 20% Q-matrix | 10% Q-matrix | 20% Q-matrix | 10% Q-matrix |
| Random | 0.1818 | 0.1818 | 0.0513 | 0.0513 | 0.0313 | 0.0313 |
| Filling Algorithm | 0.6876 | 0.6006 | 0.2483 | **0.2382** | **0.5864** | **0.4548** |
| **DCD** | **0.9292** | **0.8693** | **0.2892** | 0.1898 | 0.4943 | 0.3849 |

### A.2  Experiments under other few-labeled scenarios

We conduct experiments evaluating other missing ratios of the Q-matrix on the NIPS2020EC dataset. In Table 2 of the main body, we illustrated the results under 100%, 20% and 10% settings. Here we additionally add experiments on 50%, 30%, and 5% Q-matrix scenarios, which are illustrated in 5. The experimental results demonstrate our DCD outperforms all the interpretable baseline models in these few-labeled scenarios. Besides, as the preserved ratio of the Q-matrix decreases (from 50% to 5%), our DCD demonstrates a greater improvement (from 0.86% to 1.43%) compared to the second-ranked model in terms of DOA. This shows the advantage of DCD in label-scarce scenarios.

Table 5: Comparison in other few-labeled scenarios on NIPS2020EC dataset. We divide models into non-interpretable and interpretable models. The best scores are in bold for two kinds of models. Noted that non-interpretable models keep same results in any scenario as they do not utilize Q-matrix.

| Model | 50% Q-matrix (few-labeled) | | | | 30% Q-matrix (few-labeled) | | | | 5% Q-matrix (few-labeled) | | | |
|---|---|---|---|---|---|---|---|---|---|---|---|---|
|  | AUC↑ | ACC↑ | RMSE↓ | DOA↑ | AUC↑ | ACC↑ | RMSE↓ | DOA↑ | AUC↑ | ACC↑ | RMSE↓ | DOA↑ |
| | | | | | | NIPS2020EC | | | | | | |
| PMF | 0.814 | 0.760 | 0.407 | - | 0.814 | 0.760 | 0.407 | - | 0.814 | 0.760 | 0.407 | - |
| IRT | 0.819 | 0.762 | 0.402 | - | 0.819 | 0.762 | 0.402 | - | 0.819 | 0.762 | 0.402 | - |
| MIRT | **0.822** | **0.765** | **0.400** | - | **0.822** | **0.765** | **0.400** | - | **0.822** | **0.765** | **0.400** | - |
| DINA | 0.766 | 0.692 | 0.456 | 0.807 | 0.766 | 0.685 | 0.455 | 0.769 | 0.765 | 0.675 | 0.454 | 0.649 |
| NCDM | 0.799 | 0.739 | 0.415 | 0.802 | 0.801 | 0.743 | 0.414 | 0.765 | 0.800 | 0.742 | 0.414 | 0.644 |
| $\beta$-TCVAE | 0.794 | 0.746 | 0.416 | 0.808 | 0.795 | 0.748 | 0.412 | 0.790 | 0.801 | 0.747 | 0.415 | 0.770 |
| KSCD | 0.800 | 0.744 | 0.417 | 0.800 | 0.803 | 0.750 | 0.413 | 0.791 | 0.805 | 0.752 | 0.411 | 0.769 |
| KaNCD | 0.807 | 0.754 | 0.411 | 0.810 | 0.809 | 0.759 | 0.406 | 0.792 | 0.812 | 0.761 | 0.404 | 0.752 |
| **DCD** | **0.809** | **0.759** | **0.407** | **0.817** | **0.811** | **0.760** | **0.405** | **0.800** | **0.814** | **0.763** | **0.403** | **0.781** |

### A.3  Comparison between L1 and L2 loss for unlabeled exercises in alignment module

In section 4.3, we finally adopt the L2 loss to constrain the exercise relevance representation sparsity of unlabeled exercises. Intuitively, it's reasonable that adopt an L1 loss to constrain the value in the last part (i.e., $\boldsymbol{\mu}_{v_j}^r[min\#(K - d_1 - d_2)]$) to be 0. As shown in 6, in our initial attempt, we first directly try to use L1 loss for numerous unlabeled exercises, and we find it does not show competing performance. We speculate a possible reason is as follows: with L1 loss, after model initialization, it is very hard to revise these incorrectly labeled exercises. In contrast, L2 loss are sensitive to outliers, and would have a larger loss when the corresponding knowledge is incorrectly inferred.

Table 6: Comparison between L1 and L2 loss of alignment in 10% Q-matrix scenario on three datasets.

| Model | Matmat | | | | JunYi | | | | NIPS2020EC | | | |
|---|---|---|---|---|---|---|---|---|---|---|---|---|
| | AUC↑ | ACC↑ | RMSE↓ | DOA↑ | AUC↑ | ACC↑ | RMSE↓ | DOA↑ | AUC↑ | ACC↑ | RMSE↓ | DOA↑ |
| DCD with L1 | 0.747 | 0.851 | 0.340 | 0.777 | 0.811 | 0.770 | 0.398 | 0.634 | 0.810 | 0.758 | 0.406 | 0.722 |
| DCD with L2 | **0.764** | **0.855** | **0.334** | **0.796** | **0.814** | **0.771** | **0.397** | **0.717** | **0.813** | **0.762** | **0.404** | **0.786** |

## A.4 Monotonicity Analysis

The monotonicity assumption [38, 39] in CD works means that the probability of correct response to the exercise is monotonically increasing at any dimension of the student's knowledge proficiency, which guarantees the interpretability of student proficiency representation (i.e., the higher the value of the latent factor means the better proficiency on the corresponding knowledge concept). Given a student proficiency representation, each dimension in the representation corresponds to the student's proficiency level on a specific knowledge concept. When we only increase the value of a single dimension in the student representation (improving the student's proficiency on a specific knowledge concept), while keeping the exercise difficulty representation and relevance representation unchanged, the student's probability of answering questions related to that knowledge concept correctly will increase, while the probability of answering questions unrelated to that knowledge concept will remain unchanged. In Fig. 6, we present a case where we only increase a single dimension in the student representation, and it shows that the student's probability of answering questions related to that knowledge concept correctly increases.

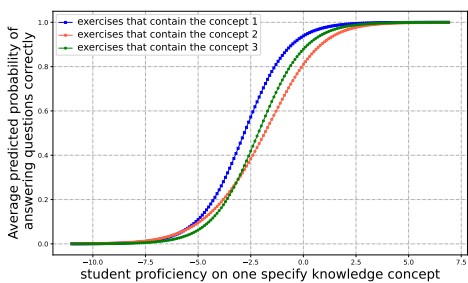

Figure 4: Example of variation of one factor in student proficiency representation. We randomly select a student and three knowledge concepts.

## A.5 Nemenyi Test

The Nemenyi test [9] is conducted to present the comparison between our proposed DCD and the interpretable baselines for all 5-fold cross-validation results. A significant difference is regarded to exist in the Nemenyi tests if the average ranks of two models differ by at least one crucial difference, which is determined using a 5% significance level. The model metric improves as the ranking score decreases. As illustrated in Fig 5, Fig 6 and Fig 7, which demonstrates the comparison on 100%, 20% and 10% labeled exercises scenario, respectively. It's obvious that our proposed DCD approach performs best for all metrics in any scenario.

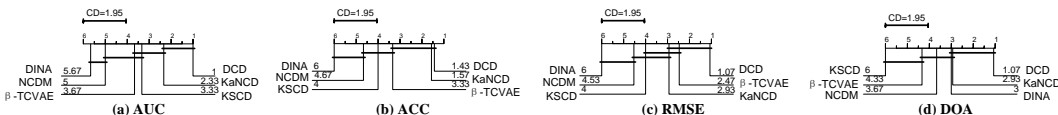

Figure 5: Nemenyi tests for comparison on 100% labeled exercises scenario

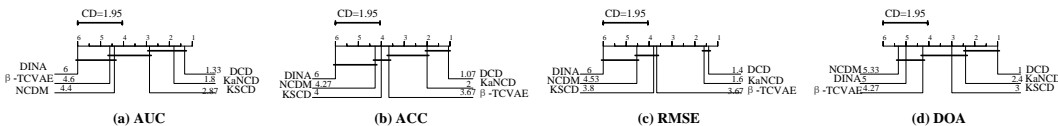

Figure 6: Nemenyi tests for comparison on 20% labeled exercises scenario

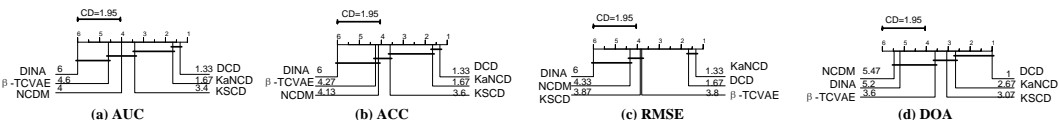

Figure 7: Nemenyi tests for comparison on 10% labeled exercises scenario

# B Experimental Details

## B.1 Reparameterization Trick

In the student encoder module, student proficiency representation $\mathbf{z}_u$ is sampled from a Gaussian distribution $\mathcal{N}(\boldsymbol{\mu}_u, \boldsymbol{\sigma}_u)$. If we sample $\mathbf{z}_u$ from $\mathcal{N}(\boldsymbol{\mu}_u, \boldsymbol{\sigma}_u)$ directly, we will lost gradient information as the sampling process is not differentiable. Fortunately, any Gaussian distribution can transform to a standard Gaussian distribution. we can sample a $\boldsymbol{\epsilon} \sim \mathcal{N}(\mathbf{0}, \mathbf{I})$ that has nothing to do with any parameters. By doing so, $\mathbf{z}_u$ is equal to $\boldsymbol{\mu}_u + \boldsymbol{\sigma}_u \odot \boldsymbol{\epsilon}$, where $\odot$ means element-wise product. However, there is no similar property of Bernoulli distribution corresponding to exercise relevance encoder. In this work, we adopt gumbel-softmax reparameterization [15] to alleviate this problem, which is one of the gradient estimator methods.

## B.2 Algorithm for Filling Missing Q-matrix

Existing interpretable baselines cannot directly apply to few-labeled scenarios. We design a filling method by employing similarity of response records in exercise to infer missing knowledge concepts. The detailed procedure is shown in Algorithm 1. We first compute the similarity (detailed in Eq. (10)) among exercises according to response records of exercise. Then for each unlabeled exercise, we vote the $vote_k$ most frequently occurring knowledge concepts among the most similar $exer_k$ labeled exercises as the filling knowledge concepts. We set $vote_k = 2$ and $exer_k = 5$ as default.

$$\mathbf{S}_{i,j} = \frac{\sum_{u_n \in \mathcal{U}} I(\mathbf{X}_{ni} \cdot \mathbf{X}_{nj} == 1)}{\sum_{u_n \in \mathcal{U}} I(\mathbf{X}_{nj} \neq 0)}, \tag{10}$$

where $\mathbf{S}_{i,j}$ denotes the similarity between exercise $v_i$ and $v_j$. $I(condition) = 1$ if $condition$ is True, and vice versa $I(condition) = 0$. $\mathbf{X}$ is the interaction matrix in training set. $\mathbf{X}_{ni} = 1, -1$ represent student $u_n$ answered exercise $v_i$ correctly and incorrectly, respectively. $\mathbf{X}_{ni} = 0$ represents student $u_n$ did not answer exercise $v_i$. $\mathcal{U}$ is the student set.

---

**Algorithm 1** Filling Missing Q-matrix for Interpretable Baselines

---

**Input:** labeled exercises set $\mathcal{V}_1$, unlabeled exercises set $\mathcal{V}_2$, interaction matrix $\mathbf{X}$ in training set, missing Q-matrix $\mathbf{Q}$, number of similar exercises $exer_k$, number of filling knowledge concepts $vote_k$
**Output:** Filling Q-matrix $\mathbf{Q}$
  1: $M = |\mathcal{V}_1 \bigcup \mathcal{V}_2|$
  2: Initialize exercise similarity matrix $\mathbf{S}$ of size $(M, M)$ with the default value $-\infty$
  3: **for all** exercise $v_i \in \mathcal{V}_1$ **do**
  4:  **for all** exercise $v_j \in \mathcal{V}_2$ **do**
  5:    Calculate $\mathbf{S}_{i,j}$ according to Eq. (10)
  6:  **end for**
  7: **end for**
  8: **for all** exercise $v_j \in \mathcal{V}_2$ **do**
  9:  $s_e$ = the most similar $exer_k$ labeled exercises of $v_j$ according to $\mathbf{S}[:, j]$
 10:  $s_k$ = the $vote_k$ knowledge concepts with the highest frequency of occurrence in $s_e$
 11:  Set corresponding elements to be one in $\mathbf{Q}[j, :]$ by $s_k$
 12: **end for**
 13: **return** $\mathbf{Q}$

---

### B.3 Datasets and Preprocessing

Our experiments are conducted on three real-world datasets, i.e., Matmat[4], Junyi[5] [4] and NIPS2020EC[6] [42], all of which contain knowledge concepts of the tree structure. For all datasets, we preserve the first-time exercise-answering record for same student-exercise pairs to support cognitive diagnosis. The detailed information on datasets and preprocessing method are depicted as following:

- **Matmat.** This dataset is collected from an intelligent web application for practicing mathematics called Matmat[7], which includes three-level knowledge concept structure after preprocessing. The first-level knowledge concept is the root node called math. The second-level knowledge concepts are organized as 5 calculational types including addition, multiplication, division and so on. The third-level knowledge concepts are organized by calculational scale.We preserve students with more than 15 response records to guarantee that each student has enough data for diagnosis.

- **Junyi.** This dataset is collected from an online educational platform called Junyi Academy[8]. There are three-level knowledge concept structure in this dataset. Besides the first-level root node, we apply area and topic field in origin dataset to denote the second-level and third-level knowledge concept ,respectively. We preserve students with more than 50 response records to guarantee that each student has enough data for diagnosis.

- **NIPS2020EC.** This dataset is originated from NeurIPS 2020 Education Challenge [42], which provides students' answers to mathematics questions from Eedi[9]. There are four-level knowledge concept structure in this dataset. We sample a subset from the task 1 of this competition by selecting response records during March 2020. We preserve students with more than 30 response records to guarantee that each student has enough data for diagnosis.

### B.4 Implementation Details

We train our model with Python 3.9 and PyTorch 1.12.1 on NVIDIA RTX A5000. The student and exercise encoders are implemented by a multilayer perceptron for all datasets. We set a prior Gaussian distribution with $\mathcal{N}(0, 1)$ for each latent factor in $\boldsymbol{\mu}_u$ and $\boldsymbol{\mu}_v^d$, and a prior Bernoulli distribution with $Bernoulli(0.2)$ for each latent factor in $\boldsymbol{\mu}_v^r$. For all interpretable models, we select the epoch with the best DOA for testing and almost set Adam as the default optimizer. The implement of DOA [38] metric is adopted from EduCDM[10]. For more implementation details, please see our public code repository `https://github.com/kervias/DCD`.

## C Knowledge Concept Tree

### C.1 Standard Knowledge Concept Tree

The employment of knowledge concept tree in our work is to obtain group based disentanglement according to granularity at different levels. For convenience, we transform the initial knowledge concept tree into a standard tree, which is defined as the tree that the height of all subtrees of any node in the tree is equal in our work. The transformation process is displayed in Fig. 8. For those leaf nodes whose depth has not reached the depth of the tree, we can continuously duplicate this node through inheritance until reaching the depth of the tree.

### C.2 Example of Real Knowledge Concept Tree

In this work, we assume that there is less independence among groups of knowledge concepts grouped by their parent nodes in the knowledge concept tree. This viewpoint is inspired by practical applications, and the analysis is as follows. To better clarify the argument, we provide the partial

---

[4]`https://github.com/adaptive-learning/matmat-web`

[5]`https://pslcdatashop.web.cmu.edu/DatasetInfo?datasetId=1198`

[6]`https://eedi.com/projects/neurips-education-challenge`

[7]`https://matmat.cz/`

[8]`https://www.junyiacademy.org/`

[9]`https://eedi.com/`

[10]`https://github.com/bigdata-ustc/EduCDM/`

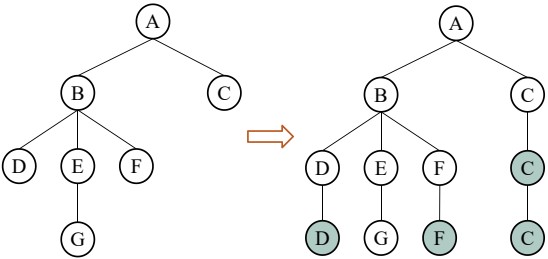

Figure 8: Process of transforming non-standard knowledge concept tree structure into standard tree.

knowledge concept tree of NIPS2020EC dataset in Fig. 9. Firstly, the course chapter hierarchy is a typical example of concepts with a tree structure. Intuitively, the associations among concepts at the high level of the hierarchy are weaker compared to the associations between concepts at the low level. For example, if we adopt the 2nd level concepts (i.e., algebra, data and statistics, ...) to group the last level concepts, the independence among groups is higher than the groups determined by the 3rd level concepts (i.e., inequalities, formula, data collection, ...). Moreover, there is a higher likelihood that concepts within the same chapter are simultaneously assessed in the same exercise. For instance, the concepts whose parent concept is inequalities would have a higher probability occur in the same exercise. However, the solving linear inequalities concept whose parent concept is inequalities and the tally charts concept whose parent concept is data collection would have a lower probability occur in the same exercise.

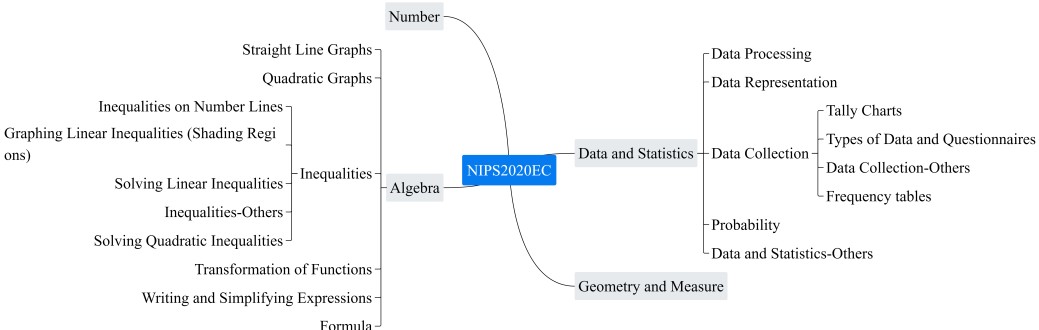

Figure 9: An example of a four-level knowledge concept tree on NIPS2020EC dataset. Here we only provide details of the hierarchical structure of *NIPS2020EC→Algebra→Inequalities* and *NIPS2020EC→Algebra→Data and Statistics→Data Collection*.

## D  Limitations

There are still some limitations of our proposed method. 1) The latent factors considered in our method are only related to knowledge concepts. However, the student answering process may refer to more factors that are not related to knowledge concepts such as the family economic conditions of students and reading comprehension difficulty of exercise textual information, which results in the incomplete and insufficient disentanglement in our proposed method. The following work we consider is to introduce more side information to help disentangle other factors. 2) The assumption that inter-group knowledge concepts are independent may be too strong. The later work we consider is to model causal relation among knowledge concepts for disentanglement. 3) The alignment module for unlabeled exercises is just to keep sparsity. In the future, we will introduce explicit relation among knowledge concepts to help infer missing knowledge concepts.

