# OpenReview forum: "Disentangling Cognitive Diagnosis with Limited Exercise Labels"
_NeurIPS.cc/2023/Conference — NeurIPS 2023 poster_

### Official Review · Reviewer_GRC1 · 2023-06-21

**Soundness:** 3 good
**Presentation:** 3 good
**Contribution:** 3 good
**Rating:** 5
**Confidence:** 5

**Summary:**

The dissociation-based cognitive diagnosis (DisenCD) model proposed in this paper addresses the cognitive diagnosis challenge of limited practice labels by using students' historical practice records to model their proficiency, practice difficulty, and practice label distribution. The model introduces novel modules to disentangle factors related to knowledge concepts and align them with the limited labels available. Experiments on three real datasets demonstrate the effectiveness of the model. Overall, this paper presents a promising approach to the cognitive diagnosis of intellectual education. However, in the process of specific description, the introduction of the two modules is too vague, and some experimental details are not provided in the article.

**Strengths:**

Practical Approach: DisenCD provides an effective approach for cognitive diagnosis under limited exercise labels, representing one of the few attempts to focus on this problem. Extensive experiments on three real-world datasets demonstrate the effectiveness of the model.
	Group-based disentanglement and limited-labeled alignment modules: The DisenCD model introduces two novel modules, group-based disentanglement, and limited-labeled alignment, which disentangle factors relevant to knowledge concepts and align them with the available limited labels. These modules help to overcome the problem of limited exercise labels and improve the accuracy of cognitive diagnosis.
	Interpretable: The DisenCD model achieves the best interpretability in all scenarios, especially in the 10% and 20% Q-matrix (label-scarce exercises) scenarios, where it significantly outperforms interpretable baselines in terms of Degree-of-Alignment (DOA). This demonstrates the superior interpretability of the model.

**Weaknesses:**

	Limited Generalizability: Although the effectiveness of the DisenCD model is demonstrated on three real-world datasets, the model's generalizability to other datasets or samples still needs to be established. Further studies are, therefore, needed to establish the generalizability of DisenCD to more extensive and diverse datasets.
	Lack of Explicit Discussion on Limitations: Although the article describes the proposed model, its training procedures, and experimental results, the limitations of the DisenCD model should be explicitly discussed. This may limit the reader's ability to interpret the results and implications of the model. A more explicit discussion of the model's limitations would have made the study more comprehensive.

**Questions:**

1.Can you please provide more details on the pre-filling algorithm used for missing Q-matrix in the experiments？

2. Could you explain further the two novel modules, group-based disentanglement and limited-labeled alignment, that DisenCD uses to disentangle relevant knowledge concepts and align them with the available limited labels?

3.Can you elaborate on the hyperparameter settings during the experiment?

**Limitations:**

No, the authors did not provide any discussions for limitations.

---

> ### Author Rebuttal · Authors · 2023-08-08
>
> We sincerely appreciate your thoughtful comments, efforts, and time. We respond to each of your questions and concerns one-by-one as follows:
>
> ### Weaknesses
> **W1-(1): Limited Generalizability: Although the effectiveness of the DisenCD model is demonstrated on three real-world datasets, the model's generalizability to other datasets or samples still needs to be established. Further studies are, therefore, needed to establish the generalizability of DisenCD to more extensive and diverse datasets.**
>
> **A1-(1)**:
> Thank you for your suggestion. We focus on proposing a cognitive diagnosis model based on a tree structure of knowledge concept relationships. Therefore, we have adopted the most popular datasets in this field, which includes the tree structure of knowledge concept relationships. In fact, these three datasets are most commonly used and representative datasets in cognitive diagnosis, given our best efforts. In the future, we would be happy to validate the generalizability of DisenCD on more datasets if we come across them.
>
>
> **W1-(2): Lack of Explicit Discussion on Limitations: Although the article describes the proposed model, its training procedures, and experimental results, the limitations of the DisenCD model should be explicitly discussed. This may limit the reader's ability to interpret the results and implications of the model. A more explicit discussion of the model's limitations would have made the study more comprehensive.**
>
> **A1-(2)**:
> Thank you. Due to the page limitation, we placed the limitations of DisenCD and an outlook on future work in the supplementary material B. We are sorry for not indicating the location of the limitation description in the manuscript. We will add it accordingly.
>
> ### Questions
>
> **Q1: Can you please provide more details on the pre-filling algorithm used for missing Q-matrix in the experiments?**
>
> **A1**:
> Thank you. Due to the page limitation, we placed this part in Section C.3 of the supplementary material in which we provide a detailed pseudo-code algorithm description on filling missing knowledge concept. Besides, we also compare DisenCD with pre-filling algorithm in section A.1 of the supplementary material.
>
> **Q2: Could you explain further the two novel modules, group-based disentanglement and limited-labeled alignment, that DisenCD uses to disentangle relevant knowledge concepts and align them with the available limited labels?**
>
> **A2**:
> Group-based disentanglement module:
>
> The fundamental assumption of disentangled representation learning is that each latent variable is independent. However, in cognitive diagnosis, each latent variable corresponds to a knowledge concept, and there are relationships between knowledge concepts. For example, certain knowledge concepts are assessed together, and one must master certain knowledge concepts before moving on to the next one. Therefore, to apply disentangled representation learning in cognitive diagnosis, we attempt to find independent relationships between knowledge concepts and group them accordingly. Knowledge concepts within a group have high correlation, while knowledge concepts between groups have low correlation, thus constraining the independence of knowledge concepts between groups. To achieve this, we incorporate knowledge concept tree structure information, where we consider higher-level knowledge concepts to have coarser granularity and higher independence among them compared to lower-level knowledge concepts.
>
> Alignment module:
>
> In the few-labeled exercise scenario, we assume that there are a large number of unlabeled exercises and a small number of labeled exercises. For the labeled exercises, we directly align dimensions with knowledge concepts in the exercise relevance representation using the supervised information from the labeled exercises. As for the unlabeled exercises, considering the sparse nature of knowledge concept annotations in the exercises, we constrain the sparsity of the unlabeled relevance. Here, we divide the exercise relevance representation into three parts: 1) Those most likely to correspond to true knowledge concepts; 2)Those serving as candidate knowledge concepts; and 3)Those with very low probability of being knowledge concepts. We use margin loss to encourage the first part to be as close to 1 as possible, allowing the model to adaptively infer missing knowledge concepts.
>
>
> **Q3: Can you elaborate on the hyperparameter settings during the experiment?**
>
> **A3**:
> Thank you. Due to the page limitation, we placed the hyperparameter settings in Section C.5 of the supplementary materials in our original submission. We will try our best to reorganize the layout to add some important information regarding the settings into the main text. Thank you again for your suggestion.

---

> > ### Comment · Reviewer_GRC1 · 2023-08-10
> >
> > I have read the rebuttal.

---

> > > ### Author Response · Authors · 2023-08-10
> > > **We are willing to discuss at any time.**
> > >
> > > We really appreciate your constructive review and your precious time. If you have any further questions or suggestions, we are very happy to discuss with you.

---

> > > > ### Comment · Reviewer_GRC1 · 2023-08-21
> > > >
> > > > I would like to thank the authors for their response. I do not have further queries at this point.

---

> > > > > ### Author Response · Authors · 2023-08-21
> > > > >
> > > > > Thank you so much for your response and valuable suggestions! We sincerely appreciate your time and efforts once again.

---

### Official Review · Reviewer_zBu1 · 2023-07-04

**Soundness:** 4 excellent
**Presentation:** 4 excellent
**Contribution:** 3 good
**Rating:** 6
**Confidence:** 5

**Summary:**

This paper presents an algorithm for cognitive diagnosis (CD) in limited data scenarios. CD aims at labeling questions to knowledge concepts.

As the CD process requires expert tagging, labelling questions to knowlege concepts in time intensive.
This paper addresses the labelling task from  three factors: student proficiency, exercise difficulty, and
exercise relevance.



**Strengths:**

 1. Knowledge concept (KC) labeling in limited data scenario
 2. Disentangled Representation Learning for interpretation
 3. Addresses correlated and independent KCs by a tree-like hierarchy


**Weaknesses:**

1. The authors utilized student interaction distribution to derive exercise relevance and knowledge concepts from exercise relevance.

One limitation of utilizing student interaction is the cold-start problem of newly introduced exercises or a small number of student responses.
A more robust way to model exercise relation is by considering the question text and student responses. See Pandey et al. [1] for reference.

2. The assumptions of inter-group and correlated KCs may be too strong.

 References.
 1. Pandey S, Srivastava J. RKT: relation-aware self-attention for knowledge tracing. In Proceedings of the 29th ACM International Conference on Information & Knowledge Management 2020 Oct 19 (pp. 1205-1214).


**Questions:**

Did the author validate (a subset of) the discovered KCs with domain experts?
Did the author evaluate how their model performs with small student responses to questions in V2?

**Limitations:**

Yes, addressed in the supplementary material.

---

> ### Author Rebuttal · Authors · 2023-08-10
>
> We sincerely appreciate your thoughtful comments, efforts, and time. We respond to each of your questions and concerns one-by-one as follows:
>
> ### Weaknesses
> **W1: The authors utilized student interaction distribution to derive exercise relevance and knowledge concepts from exercise relevance.
> One limitation of utilizing student interaction is the cold-start problem of newly introduced exercises or a small number of student responses. A more robust way to model exercise relation is by considering the question text and student responses.**
>
> **A1**:
> Thanks for your suggestions. Our algorithm focuses on inferring the exercise relevance from student responses when there is a lack of textual information. Therefore, it does rely on the quantity of interactions associated with the exercise.  In our future work, we will consider exploring the exercise relevance from both  response records and additional exercise information (e.g., text, graph, and video).
>
> **W2: The assumptions of inter-group and correlated KCs may be too strong.**
>
> **A2**: Thanks for your question. We believe that the independence among high-level concepts (coarse-grained) is higher than low-level concepts (fine-grained). This viewpoint is inspired by practical applications, and the analysis is summarized as follows. To better clarify the argument, we provide the partial knowledge concept tree of NIPS2020EC dataset in Figure 1 in our rebuttal global pdf file. Firstly, the course chapter hierarchy is a typical example of concepts with a tree structure. Intuitively, the associations among concepts at the high level of the hierarchy are weaker compared to the associations between concepts at the low level. For example, if we adopt the 2nd level concepts (i.e., algebra, data and statistics, ...) to group the last level concepts, the independence among groups is higher than the groups determined by the 3rd level concepts (i.e., inequalities, formula, data collection, ...). Moreover, there is a higher likelihood that concepts within the same chapter are simultaneously assessed in the same exercise. For instance, the concepts whose parent concept is inequalities would have a higher probability occur in the same exercise. However, the solving linear inequalities concept whose parent concept is inequalities and the tally charts concept whose parent concept is data collection would have a lower probability occur in the same exercise.
>
> ### Questions
>
> **Q1: Did the author validate (a subset of) the discovered KCs with domain experts? Did the author evaluate how their model performs with small student responses to questions in V2?**
>
> **A1**:  Thank you for your question.
>
> - Validate the discovered KCs with domain experts
>
>      Yes, we conducted experiments to validate the discovered knowledge concepts with domain experts. The results are presented in Appendix A of the supplementary materials in which we compare DisenCD with pre-filling algorithm.
>
> - Evaluate performance with small student response
>
>      We agree that the performance of the proposed algorithm rely on the quantity of interactions associated with the exercise. DisenCD may not perform  well on exercises with small student responses. As you mentioned in weakness 1, we would consider incorporating multimodal information to make up for the shortcoming.

---

> > ### Comment · Reviewer_zBu1 · 2023-08-20
> >
> > I would like to thank the authors for their response. I do not have further queries at this point.

---

> > > ### Author Response · Authors · 2023-08-21
> > >
> > > Thank you so much for your response and valuable suggestions! We sincerely appreciate your time and efforts once again.

---

### Official Review · Reviewer_B6tk · 2023-07-06

**Soundness:** 3 good
**Presentation:** 3 good
**Contribution:** 3 good
**Rating:** 7
**Confidence:** 4

**Summary:**

This paper focuses on performing cognitive diagnosis with limited exercise labels. To address the enormous cost of labeling exercises, in this paper, the authors proposed Disentanglement based Cognitive Diagnosis (DisenCD). Specifically, they first used students’ practiced records to model student proficiency, exercise difficulty, and exercise label distribution; Then, group-based disentanglement and limited-labeled alignment modules were proposed to disentangle the factors relevant to concepts and align them with real limited labels. At the same time, a tree-like structure of concepts was proposed for group-based disentangling.

**Strengths:**

The idea presented in this paper is novel and holds value in addressing the problem of limited exercise labels, as annotating them with domain experts incurs a substantial cost. Meanwhile, in the supplementary materials, it is highly significant for the authors to perform the Nemenyi Test on the experimental results.

**Weaknesses:**

1.	Figure 1: The radar chart has some mistakes. For the diagnosis of knowledge concept k2, we can find that the proficiency of student u1 is higher than that of u2, but the answer logs given by the authors show that student u1 answered v1 correctly, answered v2 incorrectly, and student u2 answered v1 correctly.
2.	In line 28, the word 'The' is written incorrectly.
3.	In line 70: MIRT has been enhanced based on IRT and can provide an assessment of subjects or students from multiple perspectives.
4.	In the experimental section:
(1)	It is clear that the model proposed by the authors did not achieve the best results in all scenarios.
(2)	Insufficient preparation of experimental parts, i.e., lack of ablation experiments to verify the effectiveness of each part.



**Questions:**

1.	In Section 2.1, The authors believed that the TextCNN method proposed by NCDM+ contained errors in extracting knowledge concepts from the semantic information of the exercise text. Could the authors provide verification of this claim? Alternatively, could the authors compare their method with the TextCNN approach to assess its effectiveness in extracting knowledge concepts?
2.	In Section 4.1, initially, the authors explained that the interaction matrix X contains significant information, including student proficiency. However, it is later mentioned that the response records of a student conceal his proficiency regarding the knowledge concepts. This raises the question of why such a contradiction exists.
3.	To the best of my knowledge, most cognitive diagnosis (CD) [1][2][3] works typically exclude students with less than 15 answer logs to ensure the plausibility of the results. However, I am curious as to why the authors chose to employ three different criteria for removing students in the three selected datasets.
4.	Regarding the 10% Q matrix or the 20% Q matrix, what criteria do the authors employ to retain 10% or 20% of the original Q matrix?

[1] Wang F, Liu Q, Chen E, et al. Neural cognitive diagnosis for intelligent education systems[C]//Proceedings of the AAAI conference on artificial intelligence. 2020, 34(04): 6153-6161.
[2] Gao W, Liu Q, Huang Z, et al. Rcd: Relation map driven cognitive diagnosis for intelligent education systems[C]//Proceedings of the 44th International ACM SIGIR Conference on Research and Development in Information Retrieval. 2021: 501-510.
[3] Ma H, Li M, Wu L, et al. Knowledge-Sensed Cognitive Diagnosis for Intelligent Education Platforms[C]//Proceedings of the 31st ACM International Conference on Information & Knowledge Management. 2022: 1451-1460.


**Limitations:**

1.	Currently, in the field of cognitive diagnosis, many datasets lack a tree structure between knowledge concepts. Therefore, the method proposed in this paper may not be suitable for the majority of datasets, i.e., the ASSISTments2009 dataset[1], and the ASSISTments2012 dataset[2].
2.	As can be seen from Table 2 in the experimental part and Table 2 in the supplementary materials, the method proposed in this paper is not well suited for datasets[1][2] with a large number of knowledge concepts. This would greatly limit the applicability of the method.

[1] https://sites.google.com/site/assistmentsdata/home/assistment-2009-2010-data/skill-builder-data-2009-2010
[2] https://sites.google.com/site/assistmentsdata/datasets/2012-13-
school-data-with-affect

---

> ### Author Rebuttal · Authors · 2023-08-10
>
> We sincerely appreciate your thoughtful comments, efforts, and time. We respond to each of your questions and concerns one-by-one as follows:
>
> ### Weaknesses
>
> **W1 and W2 : The radar chart in Figure 1 has some mistakes...., 'The', …**
>
> **A1 and A2**:
> Very sorry for the mistake and typos. In fact, the proficiency on concept k2 of student u1 should be lower than that of u2. We will update Figure 1, and correct typos in the manuscript accordingly.
>
> **W3: In line 70: MIRT has been enhanced based on IRT and can provide an assessment of subjects or students from multiple perspectives.**
>
> **A3**:
> Thank you. We agree with you that MIRT transforms the student representation from a scalar in multiple dimensions. However, in MIRT, each dimension of the multidimensional vector representing students does not correspond to a specific knowledge concept. We will rewrite this part to avoid confusion.
>
> **W4-(1): It is clear that the model proposed by the authors did not achieve the best results in all scenarios.**
>
> **A4-(1)**:
> Thank you for your comment. We compared the results of the model in terms of prediction metrics (i.e., AUC, ACC, and RMSE) and interpretability metric DOA, as shown in Table 2 of the manuscript. Only in the 10% Q-matrix scenario of the Junyi dataset, our model achieves prediction metrics comparable to the state-of-the-art model KaNCD. Moreover, for the interpretability metric DOA, the proposed model significantly outperforms the best baseline of KaNCD model. Therefore, the experimental results demonstrate the effectiveness of the proposed model.
>
>
> **W4-(2)-Brief: Insufficient preparation of experimental parts.**
>
> **A4-(2):**
> Thanks for your suggestion. We have presented the results of the ablation experiments in Figure 3 of the original submission.  Due to the page limit, we demonstrate this by hyperprameter analysis. Specifically, the disentanglement module is influenced by hyperparameter $\beta$. When $\beta$ is equal to 0, it means DisenCD without disentanglement. The margin loss and L2 loss in alignment module is influenced by $\lambda_1$ and $\lambda_2$ , respectively. When $\lambda_1$=0 and $\lambda_2$ =0 , it means DisenCD without alignment without margin loss and L2 loss, respectively. We present the ablation experiments in Table 3 in our rebuttal global pdf file. The experimental results validate the effectiveness of the disentanglement and alignment modules. We will include the separate ablation experiment results in the supplementary materials.
>
> ### Questions
>
> **Q1-Brief: Provide verification of the claim about NCDM+.... Compare DisenCD with TextCNN approach…**
>
> **A1**:
> Thanks for your question and sorry for the confusion. Extracting knowledge concepts from the exercise text is a good idea to deal with the few-labelled exercise scenario. We additionally focus on a more challenging scenario of missing exercise text scenario and infer knowledge concepts from student interaction records. According to your valuable suggestion, we would rewrite this part to avoid misunderstanding.
>
> **Q2-Brief: There exists contradiction in statement…**
>
> **A2**:
> Sorry for the confusion due to the typo. In line 154, the word “…conceal..” should be “reflect”.  We will correct this error.
>
>
> **Q3-Brief: The choice of filtering threshold for removing students.**
>
> **A3**:
> Thanks for your professional question.
>
> We conducted a survey of related literature on student filtering methods, including setting a certain filtering threshold (e.g., 15 [1], 50 [2] ) or selecting a certain number of students (e.g., 1000 [3], 10000 [4]). These methods have slight deviations in threshold settings due to different dataset sizes. Therefore, we have also designed different filtering thresholds according to the different dataset sizes. The student scales of the three datasets we used, from small to large, are Matmat, NIPS2020, and Junyi, respectively. Therefore, we set the filtering thresholds as 15, 30, and 50 in ascending order based on the student scale.
>
> Additionally, we conducted an experiment where we set the filtering threshold to 15 for NIPS2020EC dataset. The experimental results are shown in Table 1 of our global rebuttal pdf file. DisenCD outperforms the baselines, demonstrating the effectiveness of the DisenCD model.
>
> **Q4-Brief: The choice of the 10% Q matrix or the 20% Q matrix.**
>
> **A4**:
> Thanks for your question. We randomly selected two representative cases where there are 10% and 20% labeled exercises. We can also adopt other ratios.
>
> ### Limitations
> **L1-Brief:  DisenCD requiring the knowledge concept tree may not be suitable for the majority of datasets.**
>
> A1:
> Thanks for your valuable comment. Some publicly available datasets in the field of cognitive diagnosis do not include tree structure information. For such datasets, we can also constraint independence among all dimensions.
>
> More importantly, we believe that obtaining and annotating tree-structured knowledge concepts is relatively easy due to the following reasons: 1) Tree-structured knowledge concepts are typically reflected in the structural divisions of course chapters, and when designing a smart education system, chapter divisions are inevitably involved, making this data readily available. 2) Compared to cognitive diagnostic models that utilize prerequisite relationships among knowledge concepts, tree-structured knowledge concept information is easier to annotate and obtain.
>
> **L2-Brief: DisenCD is not well suited for datasets with a large number of knowledge concepts.**
>
> **A2**:
> Thanks for your insightful comment. We agree that our model may not be well suited for datasets with a large number of knowledge concepts. This is due to the Beta-TCVAE that DisenCD rely on for disentanglement, which has a high computational complexity when calculating total correlation. In the future, we will further explore the complexity issue of the model to alleviate this limitation.

---

> > ### Comment · Reviewer_B6tk · 2023-08-12
> >
> > I have read the author's response and the comments from other reviewers. Most of my concerns have been addressed. I have revised my score accordingly. Moreover, regarding to question 4, can you conduct an experiment evaluating different missing ratios of the Q matrix? Based on your response, I will determine the final score.

---

> > > ### Author Response · Authors · 2023-08-14
> > > **Experiments on other different missing ratios of the Q-matrix**
> > >
> > > We appreciate your positive comment and precious time. Regarding to question 4,  we conduct experiments evaluating different missing ratios of the Q-matrix on the NIPS2020EC dataset.  As IRT、MIRT and PMF are not affected by ratio of Q-matrix, we only  compared the interpretable models. In Table 2 of our submitted paper, we illustrated the results under 100%, 20% and 10% settings. Here we additionally add experiments on 50%, 30%, and 5% Q-matrix scenarios , which are illustrated in Table 1, Table 2, and Table 3, respectively. The experimental results demonstrate our DisenCD outperform all the interpretable baseline models in these few-labeled scenarios. Besides, as the preserved ratio of the Q-matrix decreases (from 50% to 5%), our DisenCD demonstrates a greater improvement (from 0.86% to 1.43%) compared to the second-ranked model in terms of DOA. This shows the advantage of DisenCD in label-scarce scenarios. Thank you for your suggestion, we will add the experimental results of the other scenarios to the supplementary material.
> > >
> > > \
> > > Table 1. Comparison in 50% Q-matrix scenario on NIPS2020EC dataset.
> > > | Model             | AUC↑      | ACC↑      | RMSE↓     | DOA↑      |
> > > | ----------------- | --------- | --------- | --------- | --------- |
> > > | DINA              | 0.766     | 0.692     | 0.456     | 0.807     |
> > > | NCDM              | 0.799     | 0.739     | 0.415     | 0.802     |
> > > | $\beta$-TCVAE     | 0.794     | 0.746     | 0.416     | 0.808     |
> > > | KSCD              | 0.800     | 0.744     | 0.417     | 0.800     |
> > > | KaNCD             | 0.807     | 0.754     | 0.411     | 0.810     |
> > > | **DisenCD(ours)** | **0.809** | **0.759** | **0.407** | **0.817** |
> > >
> > > \
> > > Table 2.  Comparison in 30% Q-matrix scenario on NIPS2020EC dataset.
> > >
> > > | Model             | AUC↑      | ACC↑      | RMSE↓     | DOA↑      |
> > > | ----------------- | --------- | --------- | --------- | --------- |
> > > | DINA              | 0.766     | 0.685     | 0.455     | 0.769     |
> > > | NCDM              | 0.801     | 0.743     | 0.414     | 0.765     |
> > > | $\beta$-TCVAE     | 0.795     | 0.748     | 0.412     | 0.790     |
> > > | KSCD              | 0.803     | 0.750     | 0.413     | 0.791     |
> > > | KaNCD             | 0.809     | 0.759     | 0.406     | 0.792     |
> > > | **DisenCD(ours)** | **0.811** | **0.760** | **0.405** | **0.800** |
> > >
> > > \
> > > Table 3.  Comparison in 5% Q-matrix scenario on NIPS2020EC dataset.
> > >
> > > | Model             | AUC↑      | ACC↑      | RMSE↓     | DOA↑      |
> > > | ----------------- | --------- | --------- | --------- | --------- |
> > > | DINA              | 0.765     | 0.675     | 0.454     | 0.649     |
> > > | NCDM              | 0.800     | 0.742     | 0.414     | 0.644     |
> > > | $\beta$-TCVAE     | 0.801     | 0.747     | 0.415     | 0.770     |
> > > | KSCD              | 0.805     | 0.752     | 0.411     | 0.769     |
> > > | KaNCD             | 0.812     | 0.761     | 0.404     | 0.752     |
> > > | **DisenCD(ours)** | **0.814** | **0.763** | **0.403** | **0.781** |

---

> > > > ### Comment · Reviewer_B6tk · 2023-08-17
> > > >
> > > > Thanks for your response. I have not further questions.

---

> > > > > ### Author Response · Authors · 2023-08-17
> > > > >
> > > > > Thank you very much for your valuable suggestions and  prompt response. We sincerely appreciate your time and effort once again.

---

### Official Review · Reviewer_xiTN · 2023-07-06

**Soundness:** 3 good
**Presentation:** 3 good
**Contribution:** 3 good
**Rating:** 7
**Confidence:** 3

**Summary:**

This paper introduces an innovative approach called DisenCD, aimed at enhancing the performance of cognitive diagnosis when exercise labels are limited. The proposed method incorporates two newly developed modules: the group-based disentanglement module and the limited-labeled alignment module. These modules effectively identify relevant concept factors and align them with existing labels. By leveraging limited exercise labels, the DisenCD method maximizes their potential for improved cognitive diagnosis.

**Strengths:**

1. The paper addresses an intriguing and relevant problem: cognitive diagnosis with limited exercise labels, which has practical implications.
2. The paper effectively utilizes a tree-structure to disentangle concept-related factors and align them with existing labels, providing a valuable solution to the aforementioned problem.
3. The extensive experiments conducted in this paper convincingly showcase the effectiveness of the proposed method, further strengthening its validity.

**Weaknesses:**

1. Although DisenCD aims to enhance interpretability in cognitive diagnosis, the model itself may lack interpretability. The inclusion of disentanglement and alignment modules could introduce additional complexity, making it challenging to comprehend the model's decision-making process.
2. In Section 4.2, the authors claim that there is reduced independence among knowledge concept groups in high-level tree nodes. However, the absence of necessary experiments or references weakens the support for this argument.
3. Section 5.1 lacks the necessary details regarding the implementation, which leaves gaps in understanding the practical aspects of applying the proposed method.

**Questions:**

Since the tree structure of knowledge concepts is very important for the proposed method, The details of the tree structure in each dataset should be provided.

**Limitations:**

1. DisenCD is a complex model that involves multiple modules and hyperparameters. This complexity may make it difficult to implement and tune, especially for practitioners who are not familiar with the underlying techniques.
2. DisenCD may not be scalable to large datasets due to its complexity. This may limit its applicability in real-world scenarios where large-scale cognitive diagnosis is required.

---

> ### Author Rebuttal · Authors · 2023-08-10
>
> We sincerely appreciate your thoughtful comments, efforts, and time. We respond to each of your questions and concerns one-by-one as follows:
>
> ### Weaknesses
> **W1-Brief: Although DisenCD aims to enhance interpretability in cognitive diagnosis, the model itself may lack interpretability.**
>
> **A1**: Thanks for your comment. In cognitive diagnosis, "Interpretability" means that the each dimension of learned student proficiency and exercise difficulty/relevance representation should correspond to a specific knowledge concept. Thus, it can be used to analyze the student proficiency. However, it will become difficult when there are few labeled exercises, which is the focus of our paper. We design the disentanglement and alignment modules to construct DisenCD, such that each dimension responds to a specific concept, and the variation of each dimension can control the prediction results of corresponding concepts. As such, the requirements of cognitive diagnosis can be satisfied in few-labeled exercise scenario.  This is the reason why we claimed that our proposed DisenCD is able to enhance interpretability in cognitive diagnosis.
>
> Moreover, we can understand the model's decision-making process by drawing analogies to the encoder and decoder processes in VAE-based methods. The decision process of the model involves two main stages: 1) In the encoder stage, the disentanglement module enforces the independence of the joint distribution of latent variables within each group of representations, resulting in student proficiency, exercise difficulty, and exercise relevance. The alignment module then ensures that each dimension of the representation aligns with a specific knowledge concept, giving it practical meaning. 2) In the decoder stage, prediction scores are generated based on the aforementioned three representations, similar to existing prediction methods in cognitive diagnosis.
>
> We will further clarify the model's decision-making process in the manuscript.
>
> **W2: In Section 4.2, the authors claim that there is reduced independence among knowledge concept groups in high-level tree nodes. However, the absence of necessary experiments or references weakens the support for this argument.**
>
> **A2**:
> Thanks for your question. We believe that the independence among high-level concepts (coarse-grained) is higher than low-level concepts (fine-grained). This viewpoint is inspired by practical applications, and the analysis is as follows. To better clarify the argument, we provide the partial knowledge concept tree of NIPS2020EC dataset in Figure 1 in the rebuttal global pdf . Firstly, the course chapter hierarchy is a typical example of concepts with a tree structure. Intuitively, the associations among concepts at the high level of the hierarchy are weaker compared to the associations between concepts at the low level. For example, if we adopt the 2nd level concepts (i.e., algebra, data and statistics, ...) to group the last level concepts, the independence among groups is higher than the groups determined by the 3rd level concepts (i.e., inequalities, formula, data collection, ...). Moreover, there is a higher likelihood that concepts within the same chapter are simultaneously assessed in the same exercise. For instance, the concepts whose parent concept is inequalities would have a higher probability occur in the same exercise. However, the solving linear inequalities concept whose parent concept is inequalities and the tally charts concept whose parent concept is data collection would have a lower probability occur in the same exercise.  Based on your valuable suggestions, in the revised version, we would add the group-based structure to make it clearer.
>
> **W3-Brief: Section 5.1 lacks the necessary details regarding the implementation.**
>
> **A3**:
> Thank you. We presented the implementation details in the supplementary materials, specifically in sections C.4 and C.5. Besides, we also provide the code including baselines in the supplementary material zip file. Based on your valuable suggestion, we would add a link to the codes in the main body to facilitate easy reimplementation.
>
> ### Questions
> **Q1-Brief: The details of the tree structure in each dataset should be provided.**
>
> **A1**:
> Thank you for your valuable comment. According to your suggestion, we add the partial tree structure for NIPS2020EC dataset, as shown in Figure 1 in the rebuttal global pdf file. Displaying the tree structure on these datasets helps readers to have an intuitive understanding of the operation process of DisenCD. We will update the relevant content of all datasets in the supplementary materials accordingly.
>
> ### Limitations
>
> **L1 & L2 - Brief: DisenCD is a complex model that involves multiple modules and hyperparameters. This complexity may make it difficult to implement and tune. DisenCD may not be scalable to large datasets due to its complexity.**
>
> **Answer**:
> Thanks for your comment. We agree with that the DisenCD involves multiple modules and hyperparameters and introduces additional time complexity. In this paper, we aim to design a cognitive diagnosis for numerous unlabeled exercises scenario, which significantly reduces the cost of expert annotation. Therefore, we believe that the increased complexity is acceptable compared to the huge expert annotation cost.Due to the task scenario becoming more complex and challenging, it is inevitable for the model's complexity to increase and the number of hyperparameters to grow. Besides, we have provided the code including baselines in the supplementary material zip file. If the paper is luckily to be accepted, we promise to release our project to facilitate reproducible research.  In the future, we will explore methods with lower complexity in this scenario. This is a promising direction, and we very much appreciate your suggestion.

---

> > ### Comment · Reviewer_xiTN · 2023-08-17
> >
> > The inclusion of the disentangled explanation in the supplementary material is commendable. I recommend that the author incorporate this section into the main body of the revised paper. While I am inclined to give a higher score based on this addition, I believe it's crucial to consider the feedback from other reviewers as well. I have no further questions.

---

> > > ### Author Response · Authors · 2023-08-17
> > >
> > > Thank you very much for your response. According to your suggestion, we will incorporate the disentangled explanation into the main body of the revised paper. We sincerely appreciate your professional review once again.

---

### Official Review · Reviewer_d3Wh · 2023-07-06

**Soundness:** 3 good
**Presentation:** 2 fair
**Contribution:** 3 good
**Rating:** 6
**Confidence:** 4

**Summary:**

The paper proposes a method to perform cognitive diagnosis. The method leverages notions of disentanglement representation learning to achieve better interpretability while avoiding sacrificing performance of predicting the students' answers. Experiments were conducted on three popular datasets and quantitative empirical evidence appears to support the above claim (no loss of predictive performance; improved interpretability).

**Strengths:**

The idea of leveraging disentangled representation learning in the context of cognitive diagnosis seems new.

The execution of this idea is interesting as the authors made a few changes to solve the technical challenges of applying disentangled representation learning to modeling students' skills and predict their answer correctness.

The experiments cover three popular datasets and seem comprehensive, including numerical results comparing multiple baselines and various analyses.

**Weaknesses:**

My main concern is on the notion of "interpretability". After reading the paper, I fail to understand what is "interpretability" in the authors' context. For disentangled representation learning in the vision domain, interpretability mostly means disentangling factors of variation, such that when you perform generation, you can manipulate the encoded latent vector (e.g., change a specific dimension of the vector that corresponds to one variation such as shape, size, viewpoint, etc.) to control the generation to be of a particular shape, size, viewpoint, and so on.

The authors seem to apply a similar notion of "interpretability" on the latent student knowledge/skill vector. However, I do not see any evidence or discussion on how such interpretability is manifested, e.g., what are the "disentangled factors" that the model learns? What do these factors mean and how do they connect to the knowledge concepts in the Q matrix? Do the model learn different disentanglement for different students? I believe an illustrative example would be extremely useful. Reporting only the DOA metric is not very convincing to me as a measure of "interpretability".

Because the paper makes a big claim on "interpretability" and the proposed methodology revolves around it, I am unsatisfied with the empirical evidence and analyses around it. I am open to raise my score if the authors can provide more in-depth discussion and definition around interpretability in their context, as well as illustrative examples to demonstrate the interpretability that their model learns.


----
post-rebuttal update: authors answered most of my questions and I have updated my scores accordingly.

**Questions:**

What do the authors mean by interpretability? Does it mean disentangling factors of variation on the students' latent vector? If so, what is being disentangled? Does the different dimension of the students' latent vector mean anything, or correspond to one or few concepts defined in the Q matrix?

In line 228, what's the rationale to use L2 loss? From my understanding, this loss aims to encourage sparsity, so why not using an L1 loss?

---

> ### Author Rebuttal · Authors · 2023-08-10
>
> We sincerely appreciate your thoughtful comments, efforts, and time. We respond to each of your questions and concerns one-by-one as follows:
>
> ### Questions
>
> **Q1 & Weakness - Brief: Confusion on interpretability and DOA.**
>
> **A1**: Thanks for your question. In cognitive diagnosis, different dimensions of the students' latent vector are required to correspond to specific knowledge concepts. In few-labeled exercise scenario, it is hard to meet the interpretability requirement of cognitive diagnosis. Therefore, we propose a novel DisenCD to enchance this type of interpretability through Disentangled Representation Learning (DRL). In DisenCD, interpretablity also means disentangling factors of variation. As shown in Figure 2 in the rebuttal global PDF, we randomly select three concepts, we observe that increasing the value of the correpoding concept, all students' predicted ratings on the exercises with the corresponding concpet would increase.
>
> To better illustrate the interpretability of our DisenCD, we will elaborate from two perspectives: the data generation process and the variation of disentangled factors. we will draw an analogy on DRL between the visual domain and cognitive diagnosis, and answer your questions during the process.
>
> 1.Vision Domain
>
> (1) The data generation process
>
> We can view the entire image dataset as evolving from a series of generative factors (such as color, shape, size,viewpoint, etc). For a single image, it is determined by the specific states of all the generative factors.
>
> (2) The variation of disentangled factors
>
> Given a disentangled representation, changing the value of one dimension, such as color, will result in a change in the color of the image only.
>
> 2.Cognitive Diagnosis
>
> (1) The data generation process
>
> We can view the entire response records as evolving from three groups of generative factors: student proficiency factor on each knowledge concept ($\mu_u$), exercise difficulty factor on each knowledge concept ($\mu_v^d$), and exercise relevance factor on each knowledge concept ($\mu_v^r$).
> For a response record, result of the student response to exercise $v$ depends on factors: knowledge concepts contained in the exercise $v$ (corresponding to exercise relevance factors), exercise difficulty on each knowledge concept contained in the exercise $v$ (corresponding to exercise difficulty factors) and student proficiency on each knowledge concept contained in the exercise $v$ (corresponding to student proficiency factors).
>
> (2) The variation of disentangled factors
>
> - The variation of disentangled factors in student proficiency representation.
> Given a student proficiency representation, each dimension in the representation corresponds to the student's proficiency level on a specific knowledge concept. The higher the value of a dimension, the higher the proficiency level on that knowledge concept.
> When we only increase the value of a single dimension in the student representation (improving the student's proficiency on a specific knowledge concept), while keeping the exercise difficulty representation and relevance representation unchanged, the student's probability of answering questions related to that knowledge concept correctly will increase, while the probability of answering questions unrelated to that knowledge concept will remain unchanged.
> In Figure 2 in the rebuttal global pdf file, we present a case where we only increase a single dimension in the student representation, and it shows that the student's probability of answering questions related to that knowledge concept correctly increases.
> - The variations of disentangled factors in exercise difficulty representation disentangled factors in exercise relevance representation have similar phenomenon.
>
> As for the DOA metric, the interpretability of cognitive diagnosis primarily aims to reflect the student's proficiency in various knowledge concepts when providing prediction results. Therefore, diagnostic models should provide reasonable diagnostic reports based on the student's historical response data. To meet this requirement, DOA metric is proposed to measure the degree of agreement between diagnostic report (i.e. student proficicency representation) and the whole students' historical response data. In other words, DOA metric currently is the most commonly used metric to measure the interpretability in cognitive diagnosis[1,2,4]. Perhaps this type of metric is deficient. We are also eager for more researchers to participate in cognitive diagnosis and come up with more interpretability metrics. Thanks for your comment sincerely.
>
> **Q2: In line 228, what's the rationale to use L2 loss? From my understanding, this loss aims to encourage sparsity, so why not using an L1 loss?**
>
> **A2**: Thank you for your question. In our preliminary attempt, we first directly try to use L1 loss for numerous unlabeled exercises, and we find it does not show competing performance and it is harder to reach convergence in the training process. We speculate a possible reason is as follows: with L1 loss, after model initialization, it is very hard to revise these incorrectly labeled exercises. In contrast, L2 loss are sensitive to outliers, and would have a larger loss when the corresponding knowledge is incorrectly predicted.  We show the experimental results of L1 and L2 loss in Table 2 in the global PDF file.
>
> Besides, to ensure sparsity , we use a margin-based loss as shown in the first part of Eq.(7). In the margin-based loss, after sorting each predicted exercise encoder in a descending order, we encourage that the top-ranked concepts have a large margin compared to the least-likely concepts, i.e. ,the (d1+d2+1)-th largest value to the K-th values .
> As shown in Figue 3(h), d1+d2 are set to small values. Therefore, the margin-based loss can encourage the predicted concepts to be sparse. Sorry for the confusion again, and we would explain Eq.(7) clearer in the revised version.

---

> > ### Comment · Reviewer_d3Wh · 2023-08-20
> > **on interpretability**
> >
> > Thanks for the authors' response. One more question re interpretability:
> >
> > each dimension of the learnt student proficiency factor z correspond to a knowledge concept. Does this correspondence (a dimension of z --> a specific knowledge concept) pre-determined? Or is it learnt in an unsupervised manner similar to β-TCVAE? If the latter is the case, will the correspondence between a dimension of z and a specific knowledge concept change for different optimization runs, causing an identifiability issue?
> >
> > Apologies if answers to the above questions are obvious from the paper but I guess it does not hurt to reiterate. Thanks!

---

> > > ### Author Response · Authors · 2023-08-20
> > >
> > > Thank you very much for the professional question! The question you raised regarding the correspondence between the proficiency factor and the knowledge concept is crucial to the interpretability of DisenCD. We are sorry for not clarifying this in section 4.4 decoder module.
> > >
> > > The correspondence on student proficiency representation is not pre-determined, nor does it learn in an unsupervised manner like Beta-TCVAE.
> > > Our solution is detailed as follows.
> > >
> > > The proposed semi-supervised disentangled method DisenCD comprises three representations, namely student proficiency, exercise difficulty, and exercise relevance. They all align with real knowledge concepts.
> > > First, the semi-supervised signal from the few-labeled Q-matrix would keep the correspondence on exercise relevance representation, which is detailed in the limited-labeled alignment module (Section 4.3).
> > > Then, the correspondence in exercise relevance representation ensures correspondence in both student proficiency and exercise difficulty representations through the decoder module, which is similar to existing CD works [1,2,3,4].
> > > Specifically, as shown in equation (8), the subtraction operation  $(\textbf z_{u_i} - \textbf z_{v_j}^d)$  is element-wise. The inner product operation $\otimes$ is also element-wise. The element-wise operation in the decoder module propagates the correpondence on exercise relevance representation to both student proficiency representation and exercise difficulty representation.
> > >
> > > \
> > > We are sorry for the confusion once again and will clarify this in section 4.4. Thank you!

---

> > > > ### Comment · Reviewer_d3Wh · 2023-08-20
> > > > **thanks for the clarification**
> > > >
> > > > ... and I have updated my scores accordingly.

---

> > > > > ### Author Response · Authors · 2023-08-20
> > > > >
> > > > > Thank you very much for your valuable questions! We sincerely appreciate your thoughtful comments, efforts, and time once again.

---

### Author Rebuttal · Authors · 2023-08-10

We sincerely appreciate all reviewers' thoughtful comments, efforts, and time. In the current global response, we have included an **attached rebuttal global file**. We will be **referencing some figures or tables when replying to each reviewer**. Thanks again to the reviewers for their efforts.

The references occurred in the rebuttal are listed as follows:

[1] Fei Wang, Qi Liu, Enhong Chen, Zhenya Huang, Yuying Chen, Yu Yin, Zai Huang, and Shijin Wang. Neural cognitive diagnosis for intelligent education systems. In AAAI, pages 6153–6161, 2020.

[2] Xinping Wang, Caidie Huang, Jinfang Cai, and Liangyu Chen. Using knowledge concept aggregation towards accurate cognitive diagnosis. In CIKM, pages 2010–2019, 2021.

[3] Haiping Ma, Manwei Li, Le Wu, Haifeng Zhang, Yunbo Cao, Xingyi Zhang, and Xuemin Zhao. Knowledge-sensed cognitive diagnosis for intelligent education platforms. In CIKM, pages 1451–1460, 2022.

[4] Weibo Gao, Qi Liu, Zhenya Huang, Yu Yin, Haoyang Bi, Mu-Chun Wang, Jianhui Ma, Shijin Wang, and Yu Su. Rcd: Relation map driven cognitive diagnosis for intelligent education systems. In SIGIR, pages 501–510, 2021.

---

### Decision · Program_Chairs · 2023-09-21

**Decision:**

Accept (poster)

**Comment:**

The paper tackles the problem of cognitive diagnosis, i.e., measuring students' proficiency, when limited exercises are labeled with concepts. The problem is well-motivated and many real-world settings face the challenge of limited exercise labels, given the huge labeling cost involved. The proposed method, Disentanglement-based Cognitive Diagnosis (DisenCD), addresses this problem using a semi-supervised disentanglement modeling approach. Extensive experimental evaluation on three real-world benchmarks showcases the effectiveness of the proposed method. The reviewers acknowledged that the paper considers an important problem setting and that the proposed method of cognitive diagnosis with limited exercise labels is of practical interest. However, the reviewers also raised several concerns and questions in their initial reviews. We thank the authors for their detailed responses and for actively engaging with the reviewers during the discussion phase. The reviewers have an overall positive assessment of the paper, and there is a consensus for acceptance. The reviewers have provided detailed feedback in their reviews, and we strongly encourage the authors to incorporate this feedback when preparing the final version of the paper.